# Characterizing a psychiatric symptom dimension related to deficits in goal-directed control

**Claire M Gillan[1,2,3]\*, Michal Kosinski[4], Robert Whelan[5], Elizabeth A Phelps[1,6,7], Nathaniel D Daw[8,9]**

[1]Department of Psychology, New York University, New York, United States; [2]Department of Psychology, University of Cambridge, Cambridge, United Kingdom; [3]Behavioural and Clinical Neuroscience Institute, University of Cambridge, Cambridge, United Kingdom; [4]Stanford Graduate School of Business, Stanford University, Stanford, United States; [5]Department of Psychology, University College Dublin, Dulbin, Ireland; [6]Center for Neural Science, New York University, New York, United States; [7]Nathan Kline Institute, New York, United States; [8]Department of Psychology, Princeton University, Princeton, United States; [9]Neuroscience Institute, Princeton University, Princeton, United States

**Abstract** Prominent theories suggest that compulsive behaviors, characteristic of obsessive-compulsive disorder and addiction, are driven by shared deficits in goal-directed control, which confers vulnerability for developing rigid habits. However, recent studies have shown that deficient goal-directed control accompanies several disorders, including those without an obvious compulsive element. Reasoning that this lack of clinical specificity might reflect broader issues with psychiatric diagnostic categories, we investigated whether a dimensional approach would better delineate the clinical manifestations of goal-directed deficits. Using large-scale online assessment of psychiatric symptoms and neurocognitive performance in two independent general-population samples, we found that deficits in goal-directed control were most strongly associated with a symptom dimension comprising compulsive behavior and intrusive thought. This association was highly specific when compared to other non-compulsive aspects of psychopathology. These data showcase a powerful new methodology and highlight the potential of a dimensional, biologically-grounded approach to psychiatry research.

**\*For correspondence:** claire.gillan@gmail.com

**Competing interests:** The authors declare that no competing interests exist.

## Introduction

Compulsivity is a theoretical clinical phenomenon that reflects the loss of control over repetitive self-deleterious behavior seen in a range of disorders, most notably obsessive-compulsive disorder (OCD) and addiction (*Everitt and Robbins, 2005*; *Gillan and Robbins, 2014*). But what are the underlying neural, computational, or psychological mechanisms? Researchers have suggested that compulsivity in these disorders may be partially explained by an imbalance between two different modes of control, which are more and less flexible (*Everitt and Robbins, 2005*; *Graybiel and Rauch, 2000*). In particular, a deficit in deliberative, 'goal-directed' control may leave individuals vulnerable to rely excessively on forming more rigid habits. Habits are behaviors that animals and humans learn to execute automatically when presented with familiar environmental cues (*Dickinson, 1985*). While habits are typically very useful, allowing us to efficiently perform routine actions while expending minimal cognitive effort, they cannot adapt flexibly to new situations. To override

**eLife digest** When an individual resists the temptation to stay out late in order to get a good night's sleep, he or she is exercising what is known as "goal-directed control". This kind of control allows individuals to regulate their behaviour in a deliberate manner. It is thought that a reduction in goal-directed control may be linked to compulsiveness or compulsivity, a psychological trait that involves excessive repetition of thoughts or actions. Furthermore, evidence shows that goal-directed control is reduced in people with compulsive disorders, such as obsessive-compulsive disorder (or OCD) and drug addiction. However, failures of goal-directed control have also been reported in other mental health conditions that are not linked to compulsivity, such as social anxiety disorder.

The fact that reduced goal-directed control is found across various mental health conditions highlights a core issue in modern psychiatric research and treatment. Mental health conditions are typically defined and diagnosed by their clinical symptoms, not by their underlying psychological traits or biological abnormalities. This makes it difficult to determine the cause of a specific disorder, as its symptoms are often rooted in the same psychological and biological traits seen in other mental health conditions.

To start to tackle this issue, Gillan et al. used a strategy that allowed them to look at compulsivity as a "trans-diagnostic dimension"; that is, as something that exists on a spectrum and is not specific to one disorder but involved in numerous different mental health conditions. Nearly 2,000 people completed an online task that assessed goal-directed control, and filled in questionnaires that measured symptoms of various mental health conditions. Gillan et al. showed that, as expected, people with reduced goal-directed control were generally more compulsive, and that this relationship could be seen in the context of both OCD and other compulsive disorders such as addiction.

Further, by leveraging the efficiency of online data collection to collect such a large sample, Gillan et al. were also able to examine how much different symptoms co-occurred in people. This enabled them to use a statistical technique to pick out three trans-diagnostic dimensions – compulsive behaviour and intrusive thought, anxious-depression and social withdrawal – and found that only the compulsive factor was associated with reduced goal-directed control. In fact, reduced goal-directed control was found to be more closely related to compulsivity than the symptoms of traditional mental health disorders including OCD. These findings show that research into the causes of mental health conditions and perhaps ultimately diagnosis and treatment – all of which have traditionally approached specific disorders in isolation – would benefit greatly from a trans-diagnostic approach.

our habits, organisms are capable of 'goal-directed behavior'. This refers our ability to make more considered choices, reflecting both (i) knowledge of the outcomes that our actions typically produce and (ii) our current motivation for those outcomes (*Dickinson and Balleine, 1994*). Consistent with the hypothesis that compulsion is linked to an imbalance between these modes of control, deficits in goal-directed learning have been observed across a range of putatively compulsive disorders such as drug addiction (*Sjoerds et al., 2013*; *Voon et al., 2015*), obsessive-compulsive disorder (OCD) (*Voon et al., 2015*; *Gillan et al., 2011*; *2014a*; *2014b*; *2015a*) and also binge-eating disorder (*Voon et al., 2015*). These deficits in goal-directed control have been linked to abnormal structure and function of the caudate and medial orbitofrontal cortex (*Voon et al., 2015*; *Gillan et al., 2015a*), suggesting that they may be a promising target for understanding the etiology of these disorders and thus for future treatment development.

Critically, the scope of the relationship between goal-directed learning deficits and psychopathology, and particularly their *specificity* to compulsive versus non-compulsive aspects has not been established. In fact, a similar deficit in goal-directed control was recently reported in other patient groups (*Alvares et al., 2014*; *Morris et al., 2015*), including those diagnosed with social anxiety disorder and schizophrenia, at least the former of which is not characterized by repetitive compulsive acts. This casts serious doubt over the hypothesis that goal-directed deficits are a neurocognitive mechanism that is partly responsible for psychiatric compulsivity. This lack of specificity is

unfortunately ubiquitous in psychiatry research (*Lipszyc and Schachar, 2010*; *Bickel et al., 2012*), a result, we suggest, of the broader issue that psychiatric diagnostic categories do not reflect the most discrete and neurobiologically informative phenomena. Of particular relevance to the present study are the high rates of co-morbidity between OCD and social anxiety disorder (*Ruscio et al., 2010*), the preponderance of OCD symptoms in the schizophrenia poopulation (*Poyurovsky and Koran, 2005*), and more broadly that the vast majority of patients diagnosed with obsessive-compulsive disorder (OCD) meet the criteria for another lifetime psychiatric disorder (*Ruscio et al., 2010*). Given these major overlaps, dissociating the neurocognitive bases for these respective diagnostic categories in their current form may be untenable.

Indeed, the Diagnostic and Statistical Manual of Mental Disorders (DSM), now in it's fifth edition (*American Psychiatric Association, 2013*) was developed to provide a reliable, descriptive psychiatric taxonomy, rather than an etiologically valid one. As such it is difficult to clearly discriminate the diagnostic categories it defines on the basis of genetics, neuroimaging, or indeed any of the modern tools of cognitive neuroscience. These issues have been described in detail by others (*Cuthbert and Kozak, 2013*; *Hyman, 2007*; *Robbins et al., 2012*), and have been recognized by the National Institute of Mental Health (NIMH), which has launched the Research Domain Criteria (RDoC) initiative, aiming to identify biologically plausible, trans-diagnostic markers of psychiatric disturbances (*Insel et al., 2010*). Although progress towards this goal has already been made by studies examining dissociable clusters of patients within groups diagnosed with the same disorder (*Brodersen et al., 2014*; *Fair et al., 2012*), the identification of robust, generalizable and specific markers that contribute to psychiatric co-morbidity has been curtailed by the small sample sizes that are typical of patient studies.

Accordingly, we hypothesized that a dimensional approach leveraging the efficiencies of large-scale online data collection among healthy individuals could be used to determine the precise psychiatric phenotype associated with deficits in goal-directed control, and test the specificity of this relationship with respect to other aspects of psychopathology. We hypothesized that this phenotype would broadly relate to compulsive behavior, which is seen across multiple disorders, including OCD and addiction (*Gillan et al., 2015b*), but were interested to reveal the scope and generality of this, e.g. with respect to impulsivity, a putatively related clinical phenotype (*Robbins et al., 2012*). We also wished to study how any psychiatric correlates of goal-directed control relate to variation in age and IQ, two more general factors that have been shown to covary both with goal-directed control and with some aspects of psychopathology (*Eppinger et al., 2013*; *Schad et al., 2014*; *Sandstrom et al., 1998*). To this end, rather than diagnosed patients, we used two large general-population samples collected online via Amazon's Mechanical Turk (AMT) to test (i) if compulsivity as indicated by self-report OCD symptoms is associated with individual differences in goal-directed learning, (ii) if this association generalizes to self-report symptoms of other DSM diagnostic categories that involve compulsivity, and (iii) if this association is specific to compulsive versus non-compulsive psychopathology.

Goal-directed control has recently been computationally formalized as arising from a form of reinforcement learning known as 'model-based' (*Daw et al., 2011*), which can be expressed as an individual difference measure that has been shown to predict how likely individuals are to form habits (*Gillan et al., 2015c*). Using this well-validated task (*Figure 1*) (*Daw et al., 2011*), we found support for all three postulates. In Experiment 1, we found that total scores on a self-report questionnaire measuring the severity of OCD symptoms were tracked by normal variation in model-based learning in the general population, but not by self-report anxiety or depression symptoms. In Experiment 2, we replicated the association with self-report OCD symptoms and showed that it generalized to a broader set of psychiatric symptoms that similarly involve failures in exerting control over self-deleterious behaviors, specifically alcohol addiction, eating disorders and impulsivity. Once again, we found tentative evidence for specificity with respect to non-compulsive aspects of psychopathology. Next, we conducted a factor analysis, which indicated the existence of three latent symptom dimensions that cut across the nine different questionnaires assessed in this study. Crucially, the second symptom dimension identified was characterized by 'Compulsive Behavior and Intrusive Thought', in which items were most consistently drawn from the questionnaires assessing symptoms of OCD, eating disorders and addiction, pertaining not just to repetitive compulsive behaviors (as was our prediction), but also to associated preoccupations and cognitive distortions. This factor, which was defined independently of task performance, was a significant predictor of deficits in model-based

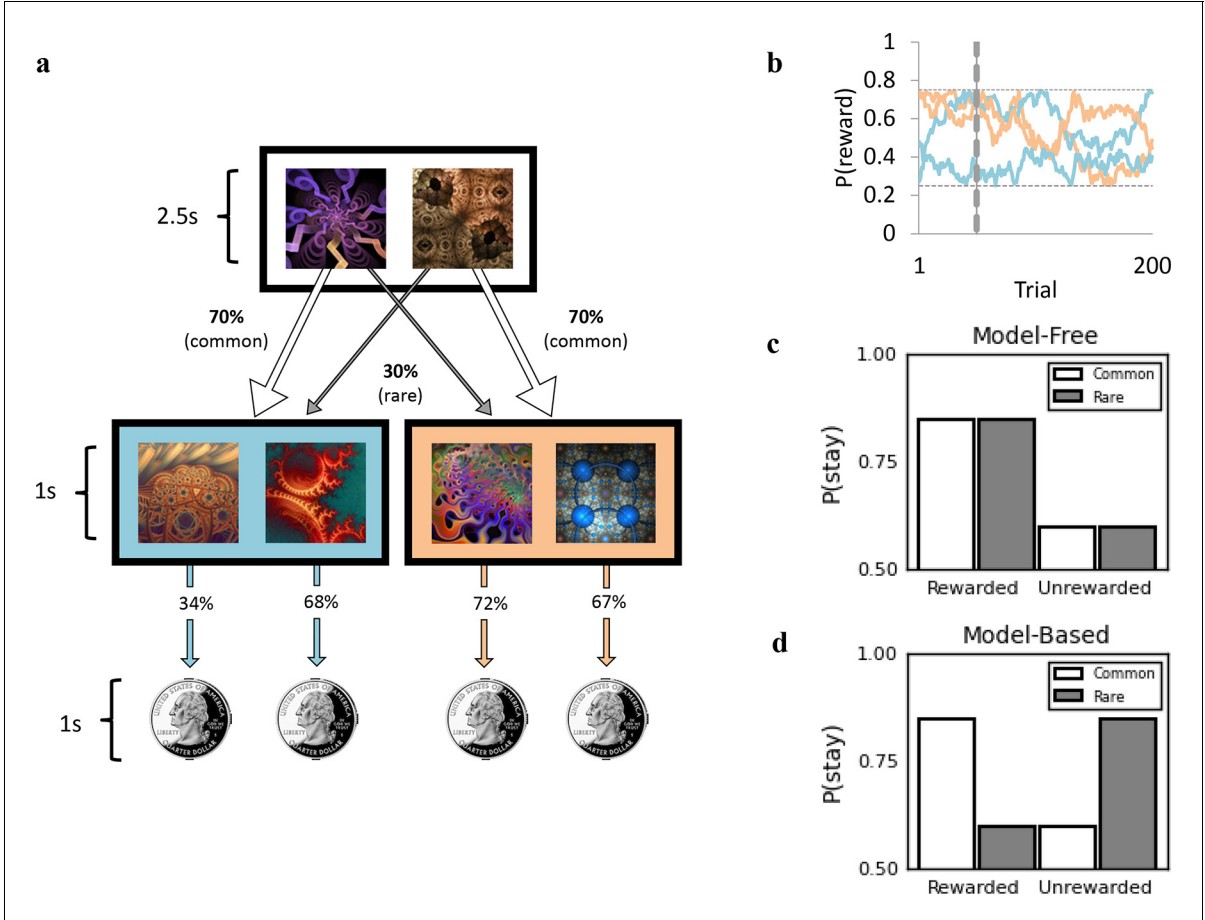

**Figure 1.** Two-step reinforcement learning task used to assess goal-directed (model-based) learning. (a) Subjects chose between two fractals, which probabilistically determined whether they would transition to the orange or blue second stage state. For example, the fractal on the left had a 70% chance of leading to the blue second stage state ('common' transition) and a 30% chance of leading to the orange state ('rare' transition). These transition probabilities were fixed and could be learned over time. In the second stage state, subjects chose between two fractals, each of which was associated with a distinct probability of being rewarded with a 25 cents coin. The probability of receiving a reward associated with each second stage fractal could also be learned, but (unlike the transition structure) these drifted slowly over time ($0.25 < P < 0.75$, panel b). This meant that in order to earn the most rewards possible, subjects had to track which second stage fractals were currently best as they changed over time. Reward probabilities depicted (34%, 68%, 72%, 67%) refer to example trial 50, denoted by the vertical dashed line in (b). (b) Drifting reward probabilities determined by Gaussian Random Walks for 200 trials with grey horizontal lines indicating boundaries at 0.25 and 0.75. (c) Schematic representing the performance of a purely 'model-free' learner, who only exhibits sensitivity to whether or not the previous trial was rewarded vs. unrewarded, and does not modify their behavior in light of the transition that preceded reward. (d) Schematic representing the performance of a purely 'model-based' learner, who is more likely to repeat an action (i.e. 'stay') following a rewarded trial, only if the transition was common. If the transition to that rewarded state was rare, they are more likely to switch on the next trial.

learning. Crucially, this effect was highly specific to this factor, when directly compared to the two other factors identified in this analysis, 'Anxious-Depression' and 'Social Withdrawal'.

## Results

In Experiment 1, we tested the hypothesis that individual differences in total scores on a questionnaire assessing the severity of OCD symptoms are associated with normal variation in goal-directed control, rather than necessitating the categorical comparison of OCD patient vs. control groups. Participants (N = 548) first completed a reinforcement-learning task that quantifies individual differences in goal-directed ('model-based') learning, which is operationalized as a parameter estimate from a logistic regression analysis predicting choices in the task (see Materials and methods and refs [*Daw et al., 2011*; *Gillan et al., 2015c*]). Next, we administered a short Intelligence Quotient (IQ)

test, followed by self-report questionnaires assessing symptoms of OCD, along with depression and trait anxiety, which we did not expect to be associated with goal-directed deficits. In line with our hypothesis, there was a significant association between scores on the OCD questionnaire and goal-directed deficits (i.e. a negative relationship between OCD severity and model-based learning; β = −0.040, Standard Error (SE) = 0.02, p=0.049) when (as in all analyses reported henceforth) controlling for age, IQ and gender, which have been previously reported to covary with goal-directed behavior (*Eppinger et al., 2013*; *Schad et al., 2014*; *Sandstrom et al., 1998*). Specifically, for each increase of 1 standard deviation (SD) in the total score on the OCD questionnaire, model-based learning was reduced by 14%. No such relationship was observed for self-report depression (β = −0.016, SE = 0.02, p=0.439) or trait anxiety (β = −0.006, SE = 0.02, p=0.777) severity (*Table 1*, *Figure 2A*). Moreover, the relationship between total scores on the OCD questionnaire and goal-directed deficits survived inclusion of the depression and trait anxiety total scores in the same model as covariates (β = −0.048, SE = 0.02, p=0.04). These data indicate that deficits in goal-directed control are a marker of normal variation in OCD symptomatology in the general population.

In Experiment 2, we aimed to test the reliability, generalizability and specificity of this finding in a larger cohort of task-naïve subjects (N = 1413, based on a power analysis given the aforementioned results). The procedure was identical to that in Experiment 1, except for the addition of several more clinical questionnaires. To test for generalizability, we assessed symptoms associated with other disorders that have been hypothesized to have compulsive features. In addition to the OCD questionnaire used in Experiment 1, these pertained to alcohol addiction, eating disorders, along with aspects of impulsivity and schizotypy (*Everitt and Robbins, 2005*; *Poyurovsky and Koran, 2005*; *Robbins et al., 2012*; *Godier and Park, 2014*). To test for specificity, in addition to the mood symptoms assessed previously (depression and trait anxiety) we added self-report measures assessing social anxiety and apathy; we also predicted that non-compulsive aspects of schizotypy and impulsivity might not be associated with goal-directed control. In this independent sample, we replicated the results from Experiment 1; scores on the OCD questionnaire were significantly associated with

**Table 1.** Self-report questionnaire total scores and model-based learning.

| Questionnaire | β (SE) | z-value | p-value |
|---|---|---|---|
| *Experiment 1 (N=548)* | | | |
| OCD | -0.040 (0.02) | -1.97 | 0.049 * |
| Depression | -0.016 (0.02) | -0.77 | 0.439 |
| Trait Anxiety | -0.006 (0.02) | -0.28 | 0.778 |
| *Experiment 2 (N=1413)* | | | |
| Eating Disorders | -0.037 (0.01) | -3.30 | <0.001 *** |
| Impulsivity | -0.034 (0.01) | -3.01 | 0.007 ** |
| OCD | -0.026 (0.01) | -2.34 | 0.020 * |
| Alcohol Addiction | -0.025 (0.01) | -2.18 | 0.029 * |
| Schizotypy | -0.017 (0.01) | -1.48 | 0.14 |
| Depression | -0.010 (0.01) | -0.87 | 0.385 |
| Trait Anxiety | -0.008 (0.01) | -0.68 | 0.498 |
| Apathy | -0.001 (0.01) | -0.06 | 0.953 |
| Social Anxiety | 0.008 (0.01) | 0.68 | 0.496 |

*p<0.05; **p<0.01; ***p<0.001.

SE=standard error.

Each row reflects the results from an independent analysis where each questionnaire total score (z-transformed) was entered as SymptomScorez in the following model: glmer(Stay ~ Reward * Transition * SymptomScorez + Reward * Transition * (IQz + Agez + Gender) + (Reward * Transition + 1 | Subject)). Model-based statistics refer to the following interaction: SymptomScorez x Reward x Transition. For each, positive β values indicate that the symptom score is associated with greater model-based learning, while negative β values indicate that the symptom score is associated with reduced model-based learning.

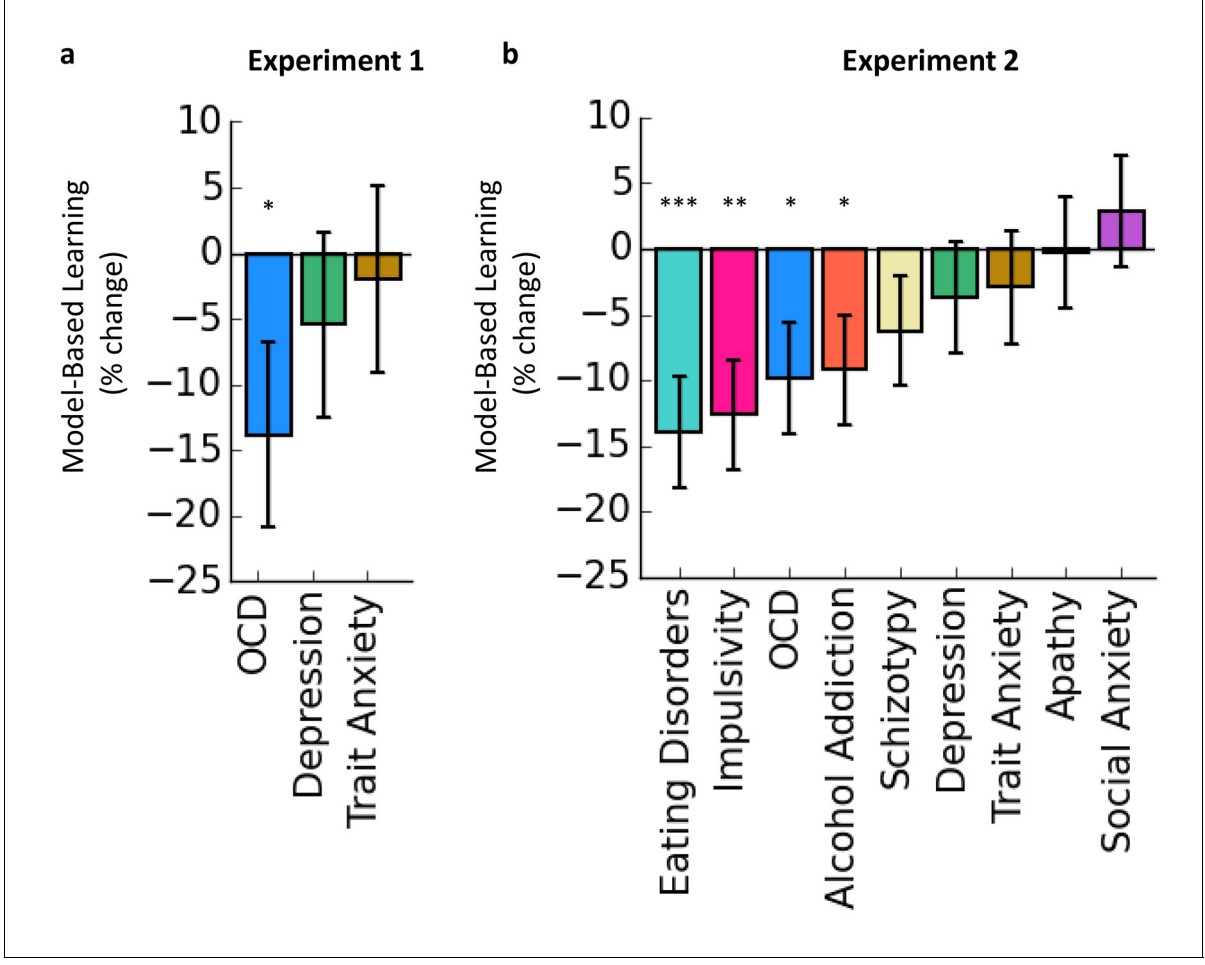

**Figure 2.** Associations between Goal-directed (model-based) deficits and self-reported psychopathology. The y-axes indicate the% change in model-based learning for each change of 1 standard deviation (SD) of clinical symptoms. Error bars denote standard error. (a) In Experiment 1, total scores on a self-report questionnaire assessing OCD symptoms in a general population sample were associated with deficits in goal-directed (model-based) learning. Specifically, for each increase of 1 SD in OCD symptoms reported, model-based learning was 14% lower than the group mean. No effects were observed in depression or trait anxiety. (b) In Experiment 2, the results from Experiment 1 were replicated: OCD symptoms were associated with deficits in goal-directed learning, while total scores on questionnaires assessing depression and trait anxiety were not. We found that the association between compulsive behavior and goal-directed deficits generalized to symptoms associated with other disorders that are similarly characterized by a loss of control over behavior, alcohol addiction, eating disorders and impulsivity. No significant effects were observed for scores on questionnaires assessing schizotypy, depression, trait anxiety, apathy or social anxiety.

goal-directed deficits ($\beta=-0.026$, SE=0.01, p=0.020), while controlling for age, gender and IQ (*Table 1*, *Figure 2B*). As we hypothesized, this effect generalized to phenotypically disparate manifestations of psychiatric compulsivity: total scores on self-report measures of eating disorder severity ($\beta=-0.037$, SE=0.01, p<0.001), impulsivity ($\beta=-0.034$, SE=0.01, p=0.007) and alcohol addiction ($\beta=-0.025$, SE=0.01, p=0.029). Also as predicted, we found no significant associations between goal-directed deficits and total scores on the depression ($\beta=-0.01$, SE=0.01, p=0.385), apathy ($\beta=-0.001$, SE=0.01, p=0.953), trait anxiety ($\beta=-0.008$, SE=0.01, p=0.498) or social anxiety ($\beta=0.008$, SE=0.01, p=0.496) questionnaires. We found no significant association between self-report levels of schizotypy and goal-directed control ($\beta=-0.017$, SE=0.01, p=0.14), possibly reflective of the great deal of heterogeneity within this particular psychiatric construct.

Previous studies using this task have assessed an individual's goal-directed learning in either of two ways: predicting their choices using either a regression model (as reported above) or the fit of a more elaborate computational learning model, which the regression model approximates. In separate analyses using fits of the computational learning model (see Materials and methods;

*Supplementary file 5A*), all of the aforementioned results were recapitulated, with the exception that the relationship between OCD and model-based learning in Experiment 1 fell short of significance (but was significant in Experiment 2) and schizotypy reached significance in Experiment 2 as a negative predictor of model-based learning.

Given both the heterogeneity within, and the high correlation across these questionnaires (e.g., Depression and Trait Anxiety scores correlate at $r=0.81$) these questionnaires, assessing the statistical specificity of these effects by including their total scores in the same model is both methodologically and conceptually fraught. To address this issue, we conducted a factor analysis based on the 209 individual questionnaire items, thereby reducing the collinearity across scores on these psychiatric questionnaires. Note that this analysis was carried out on the questionnaire scores alone, without reference to the results on the reinforcement learning task. We found evidence for three dissociable factors ('dimensions') that cut across the nine questionnaires from which items were drawn, which we labeled 'Anxious-Depression', 'Compulsive Behavior and Intrusive Thought' and 'Social Withdrawal', based on the loadings of individual items (*Supplementary file 2A–C*, *Table 2*, *Figure 3A*). Although the labeling of factors is of course a subjective process, quantitatively speaking, 'Compulsive Behavior and Intrusive Thought' had high and consistent loadings from almost all items pertaining to eating disorders (Mean loading=0.36, SD=0.15), OCD (Mean loading=0.50, SD=0.06) and addiction (Mean loading=0.31, SD=0.07), which have all been couched as 'compulsive' disorders in the literature (*Everitt and Robbins, 2005*; *Gillan and Robbins, 2014*; *Godier and Park, 2014*) (*Table 2*). In addition to picking up every self-report item that pertained to compulsive behavior from our question pool, the loadings on Factor 2 were not confined to compulsive behaviors, but equally featured items pertaining to related patterns of thought, i.e. obsessions, preoccupations, or intrusive thoughts. We cannot speak to causality here, but this suggests that repetitive behavior and repetitive, irrational patterns of thought are not orthogonal symptom dimensions, but perhaps share a common neurobiological root. Items from the impulsivity scale (of which the total score was a significant predictor of goal-directed deficits) did not load as strongly or consistently on this factor (M=0.15, SD=0.15; significantly less than the former three questionnaires, Eating Disorders vs. Impulsivity: t(52)=5.178; OCD vs. Impulsivity: t(41)=11.379; Alcohol Addiction vs. Impulsivity: t(33) =4.342, all p<0.001) (*Table 2*).

We next tested for an association between subjects' scores on these three factors and their, separately measured, goal-directed performance. When tested alone, 'Compulsive Behavior and Intrusive Thought' was significantly associated with deficits in goal-directed learning (β=−0.046, SE=0.01, p<0.001), and this effect size was greater than that of any of the questionnaires used in this study, corresponding to a 17% reduction in model-based learning for an increase of 1 SD in 'Compulsive Behavior and Intrusive Thought' (*Figure 3B*, *Table 3*). There were no significant effects of Factor 1 (β=−0.001, (0.01), p=0.92) or Factor 3 (β=0.013, SE=0.01, p=0.24) on model-based learning. Finally, we directly compared the associations between goal-directed deficits and these factors by including

**Table 2.** Means and standard deviations (in parentheses) of loadings onto Factor 1 'Anxious-Depression', Factor 2 'Compulsive Behavior and Intrusive Thought' and Factor 3 'Social Withdrawal' factors for each questionnaire.

| | Anxious-depression | Compulsive behavior and intrusive thought | Social withdrawal |
|---|---|---|---|
| | (Factor 1) | (Factor 2) | (Factor 3) |
| Alcohol addiction | 0.15 (0.05) | **0.31 (0.07)** | -0.23 (0.06) |
| Apathy | **0.44 (0.16)** | -0.05 (0.13) | 0.04 (0.13) |
| Depression | **0.38 (0.23)** | 0.14 (0.14) | 0.04 (0.06) |
| Eating disorders | -0.05 (0.10) | **0.36 (0.15)** | 0.06 (0.06) |
| Impulsivity | 0.24 (0.22) | 0.15 (0.15) | -0.11 (0.11) |
| OCD | -0.05 (0.14) | **0.50 (0.06)** | 0.09 (0.07) |
| Schizotypy | 0.16 (0.11) | 0.18 (0.13) | 0.08 (0.14) |
| Social anxiety | 0.04 (0.05) | 0.08 (0.09) | **0.57 (0.14)** |
| Trait anxiety | **0.52 (0.17)** | 0.15 (0.16) | 0.13 (0.08) |

Scores greater than 0.25 are emboldened to highlight the dominant constructs.

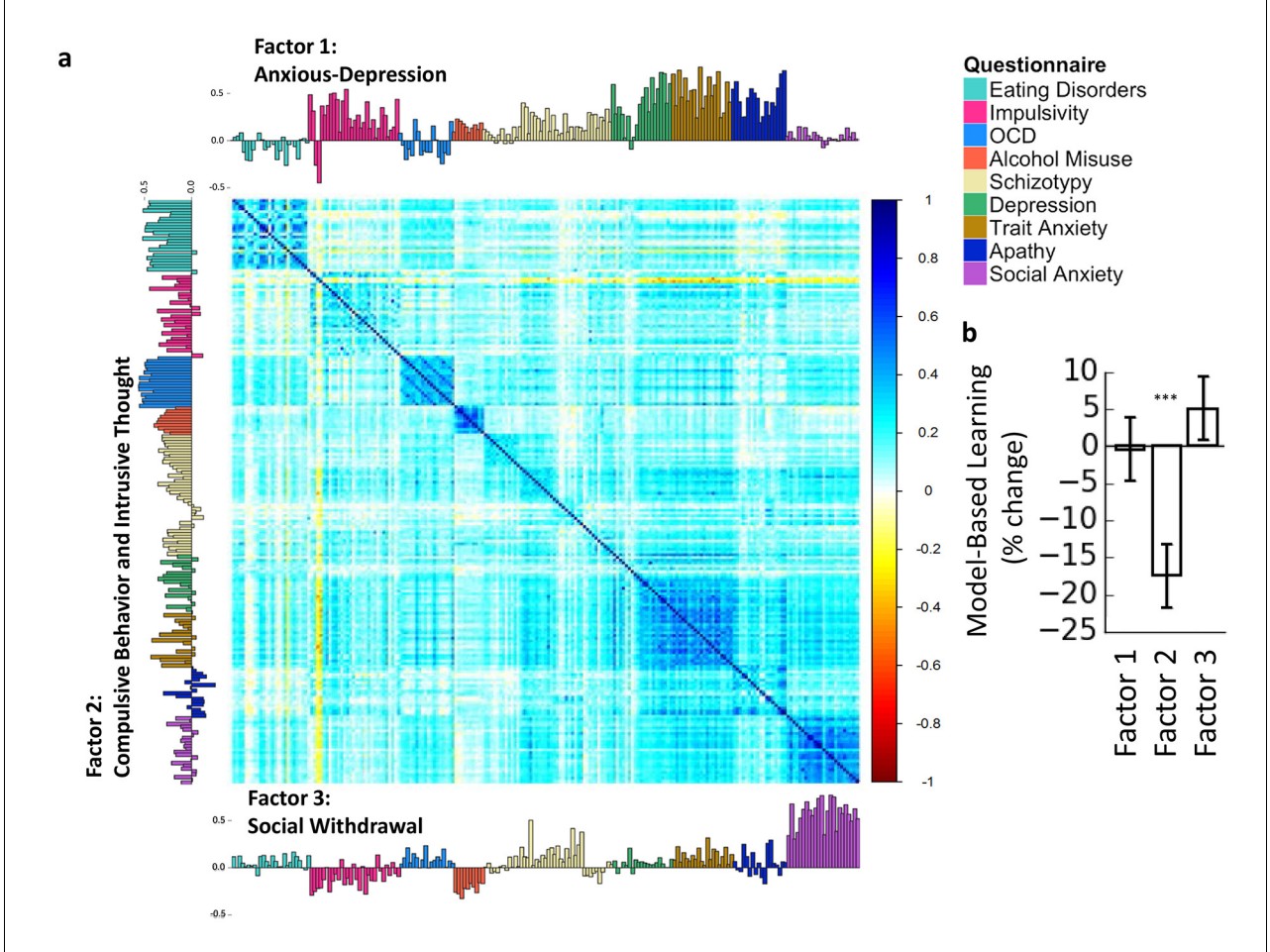

**Figure 3.** Trans-diagnostic factors. (a) Factor analysis on the correlation matrix of 209 questionnaire items suggested that 3-factor solution best explained these data. Factors were 'Anxious-Depression', 'Compulsive Behavior and Intrusive Thought' and 'Social Withdrawal'. Item loadings for each factor are presented on the top, left and bottom sides of the correlation matrix, color-codes indicate the questionnaire from which each item was drawn. (b) These factors were entered into mixed-effects models, revealing that only the Factor 2 'Compulsive Behavior and Intrusive Thought' was associated with goal-directed deficits, the effect size (17% reduction in model-based learning for every 1 SD increase in 'Compulsive Behavior and Intrusive Thought') was larger than for any individual questionnaire, and pairwise contrasts revealed that these deficits were specific to this factor, compared to Factor 1 'Anxious-Depression' and Factor 3 'Social Withdrawal'. The y-axes indicate the% change in model-based learning for each change of 1 standard deviation (SD) of clinical symptomatology. Error bars denote standard error.

them in the same model and conducting planned contrasts. We found that deficits in goal-directed control were highly specific to the 'Compulsive Behavior and Intrusive Thought' (vs. 'Anxious-Depression', β=−0.062, SE=0.02, p=0.001; vs. 'Social Withdrawal', β=−0.089, SE=0.02, p<0.001). Moreover, when included in the same model with the other factors, 'Social Withdrawal' (onto which addiction and aspects of impulsivity load negatively) emerged as a significant *positive* predictor of goal-directed control over action (β=0.031, SE=0.01, p=0.014). To test the extent to which the relationship between goal-directed deficits and 'Compulsive Behavior and Intrusive Thought' is truly continuous, we carried out a supplementary analysis in which this factor was entered as a quadratic term in our model, thereby testing for a nonlinear effect. We found no evidence for nonlinearity (beta=−0.0016, p=0.822), and the linear effect remained significant when included in this model (beta=−0.045, p=0.001). Similarly, we repeated our analyses in subsets of our population comprising either 'putative patients' (defined as those who scored in the top 25% on a given self-report measure) or subjects in the normal range (bottom 75%) and the results were broadly consistent across sub-samples (Materials and methods, *Supplementary file 3*).

**Table 3.** Trans-diagnostic factors and model-based learning.

| Construct | β (SE) | z-value | p-value |
|---|---|---|---|
| *Independent Models* | | | |
| 'Anxious-Depression' (Factor 1) | -0.001(0.01) | 0.10 | 0.920 |
| **'Compulsive Behavior and Intrusive Thought' (Factor 2)** | **-0.046(0.01)** | **-4.06** | **<0.001 \*\*\*** |
| 'Social Withdrawal' (Factor 3) | 0.013(0.01) | 1.18 | 0.238 |
| *Covariate Model* | | | |
| 'Anxious-Depression' (Factor 1) | 0.003(0.01) | 0.28 | 0.781 |
| **'Compulsive Behavior and Intrusive Thought' (Factor 2)** | **-0.058(0.01)** | **-4.71** | **<0.001 \*\*\*** |
| **'Social Withdrawal' (Factor 3)** | **0.031(0.01)** | **2.45** | **0.014\*** |

*p<0.05; **p<0.01; ***p<0.001.

SE=standard error.

Top panel shows results from Independent Models. Bottom panel shows results from Covariate Model, where trans-diagnostic factors were entered together into the same model: glmer(Stay ~ Reward * Transition * (Factor1z + Factor2z + Factor3z + IQz + Agez + Gender) + (Reward * Transition + 1 | Subject)). Statistics refer to the interaction between scores on each factor and Reward x Transition, i.e. the extent to which that score is associated with changes in model-based learning. Positive β values indicate that the symptom score is associated with greater model-based learning, while negative β values indicate that the symptom score is associated with reduced model-based learning.

Finally and in complement to the unsupervised factor analysis used to define 'Compulsive Behavior and Intrusive Thought', we carried out a fully supervised analysis (regression with elastic net regularization) to identify directly from the individual questionnaire items those most predictive of goal-directed learning, as assessed using the regression model. Supporting our previous conclusions, those items that predicted model-based deficits in the negative direction substantially overlapped with items with above-threshold loadings on 'Compulsive Behavior and Intrusive Thought' (75% overlap; *Supplementary file 4*). One noteworthy pattern arises among the exceptions. The supervised analysis also identified several additional items from the impulsivity questionnaire, which had not loaded on 'Compulsive Behavior and Intrusive Thought', but did predict goal-directed learning. Those were items that tracked subjects' motivation to engage with the experimental paradigm, e.g. "I (do not) like to think about complex problems". Other, more compulsivity-relevant items from the impulsivity scale, involving compulsive shopping and general loss of control over action, were identified in both analyses. The former items are likely of little clinical relevance, but can explain the strong association between impulsivity total scores and goal-directed deficits, despite the fact that impulsivity did not load strongly onto 'Compulsive Behavior and Intrusive Thought'.

In addition to tracking one well-delineated aspect of psychopathology, we found that task performance was significantly related to other measures collected in this study. First, although individual variation in 'model-free' performance on the learning task did not track any of the scores from our psychiatric questionnaires (*Supplementary file 1C*), in Experiment 1, model-free performance did relate significantly to age (*Supplementary file 1B*, Reward*Age interaction). 'Model-based' learning was also related to age and IQ. In particular, higher IQ was associated with increases in goal-directed, 'model-based' learning. In contrast to the effect of age on 'model-free' learning, older people were significantly less 'model-based' compared to their younger counterparts. All of these results were replicated in Experiment 2. Additionally, the larger sample size in Experiment 2 allowed us to detect small but significant associations between gender and model-free and model-based learning. Males were significantly less model-free and more model-based relative to females tested in this study. Importantly, all of these effects are controlled for (by including age, IQ, and gender as additional covariates) in the analyses relating learning to psychiatric symptoms.

## Discussion

Here, we tested the utility of a dimensional approach to investigating the neurocognitive basis of compulsivity using two large-scale general population samples. Evidence from multiple complimentary analyses supported the conclusion that 'Compulsive Behavior and Intrusive Thought' is a symptom dimension associated with deficits in goal-directed control that links features of multiple psychiatric disorders; most notably symptoms of OCD, addiction, and eating disorders. Interestingly, this dimension goes beyond the uncontrolled behaviors that have been previously associated with compulsivity, to include obsessions, preoccupations and intrusive thoughts.

That self-report scores of OCD and addiction symptoms were associated with these deficits is consistent with previous research in patient populations (*Sjoerds et al., 2013*; *Voon et al., 2015*; *Gillan et al., 2011*; *2014a*; *2014b*; *2015a*), and extends these results for the first time to a general population sample. Likewise, binge-eating disorder has also been previously associated with reduced goal-directed control in one patient study and an animal model (*Voon et al., 2015*; *Furlong et al., 2014*). Critically, the results of the present study extend this finding to self-report symptoms of other subtypes of eating disorders, suggesting that Compulsive Behavior and Intrusive Thought (and associated deficits in goal-directed control) are a key component of more aspects of eating disorders than previously documented. An entirely consistent exception was that items relating to exerting control over food intake (e.g. "I display self-control around food") did not load strongly on the 'Compulsive Behavior and Intrusive Thought' factor.

A previous study reported an association between social anxiety disorder and deficits in goal-directed control (*Alvares et al., 2014*). Using self-report social anxiety symptom scores in our general population sample, we did not replicate this finding, and in fact observed a trend towards enhanced goal-directed control associated with social anxiety symptoms. Specifically, in most analyses social anxiety symptoms (both total scores and the 'Social Withdrawal' factor) was unrelated to task performance. We did however observe a significant *positive* association between the 'Social Withdrawal' factor and goal-directed control in one analysis, while controlling for the other factors in the same analysis. This result should be interpreted with caution, given that the association was not sufficiently robust to predict goal-directed control alone, but this serves to illustrate that 'Social Withdrawal' trended towards predicting better goal-directed control, not worse. Two explanations for the discrepant findings between the present study and the prior investigation with diagnosed social anxiety disorder patients are the differences in sample size between our respective studies and that the co-morbidities reported for the social anxiety disorder population of the study by *Alvares and colleagues (2014)* could not be controlled for and may have driven the reported association. This underscores the importance of a dimensional approach to psychiatric phenotyping.

Schizophrenia has also been previously associated with deficits in goal-directed control (*Morris et al., 2015*), a finding that was partially supported by the present study (to the limited extent that 'schizotypy', measured here, has implications for schizophrenia as a clinical condition). Consistent with the heterogeneous nature of schizophrenia, where two diagnosed patients can have entirely non-overlapping symptoms (*American Psychological Association, 2013*), we did not find a significant association between the total score on the schizotypy questionnaire and deficits in goal-directed control (although this was significant in a second analysis based on a full computational model). However, using our trans-diagnostic approach, we found that in particular 'unusual experiences' characteristic of schizotypy loaded onto the 'Compulsive Behavior and Intrusive Thought' factor, which in turn was a strong predictor of goal-directed deficits. This finding converges with studies highlighting that delusions are more closely linked to executive deficits than the negative symptoms of schizophrenia (*Lysaker et al., 1998*; *2003*). In terms of clinical phenomenology, schizophrenia and OCD share a common pattern of abnormal beliefs and as DSM-5 and others have noted, the distinction between a delusion in schizophrenia and a strongly held belief in OCD is often blurred (*Poyurovsky and Koran, 2005*; *American Psychological Association, 2013*). These data suggest that 'Compulsive Behavior and Intrusive Thought', which comprises automatic behaviors as well as associated repetitive thoughts, may be common to both schizophrenia and OCD and explained by deficits in goal-directed control.

Earlier work investigating deficits in goal-directed learning in compulsive patient populations did not employ a positive clinical control (*Voon et al., 2015*), therefore until now the possibility that goal-directed deficits were non-specific, i.e. evident in all psychiatric populations, remained

untested. For instance, prior studies have found a consistent association between stress and goal-directed learning deficits (*Otto et al., 2013*; *Schwabe and Wolf, 2009*), which might in principle mediate non-specific effects due to the considerable burdens of mental illness. Here, we tested this possibility rigorously in two independent samples. We found no association between 'Anxious-Depression' and deficits in goal-directed control, and moreover the specificity of goal-directed deficits to 'Compulsive Behavior and Intrusive Thought' was confirmed through direct statistical comparisons.

Prior work has shown that the model-based learning deficits predict the presence of habits using a devaluation probe (*Gillan et al., 2015c*), providing a tentative mechanism through which the goal-directed deficits observed in the present study might cause the development of compulsive behaviors. Indeed, this converges with prior work showing that when OCD patients are performing habits, they show dysfunctional hyperactivity in the caudate (*Gillan et al., 2015a*), a region associated with goal-directed control over behavior (*Dolan and Dayan, 2013*). An outstanding question, however, is the extent to which excessive stimulus-response habit learning also contributes to Compulsive Behavior and Intrusive Thought. The model-free component of the task we employed in the present study did not relate significantly to psychiatric symptomatology, as indeed we had hypothesized because it also does not appear to be sensitive to slow habitual learning (indeed, unlike the model-based component of the task, it does not predict devaluation [*Gillan et al., 2015c*]). Future work is needed to develop a computational marker of individual differences in stimulus-response habit formation, so that this possibility can directly be tested.

Another interesting question that emerges from these data is how deficits in goal-directed control might result in both cognitive distortions (which take the form of obsessions in OCD, such as a fear of germs) and compulsive behavior (e.g. repetitive hand-washing), which our factor analysis suggested are inextricably linked. One possibility was raised by a recent study, which demonstrated that just like low-level stimulus-response behaviors, more abstract goal selection can also be rendered habitual (*Cushman and Morris, 2015*). If these habitual cognitive actions can be conceived as a sort of 'habit of thought,' this might indicate a common mechanism for both compulsive behavior and the related repetitive patterns of thought (i.e. 'habits of thought'). An alternative possibility posits that obsessive thoughts may develop as a *result* of compulsive behavior (*Gillan and Robbins, 2014*). Evidence for this idea comes from a study where OCD patients were found to engage in *post-hoc* rationalization in order to explain a series of habitual responses (*Gillan et al., 2014b*). The notion is that in OCD, experiencing a recurrent urge to wash one's hands might cause a patient to infer that they are concerned about hygiene. Future, longitudinal work will be needed to dissect the temporal dynamics of these symptom features to test these hypotheses, which are not mutually exclusive.

Researchers have suggested that 'Impulsivity' and 'Compulsivity' are partially overlapping neuro-cognitive features relevant for many psychiatric disorders (*Robbins et al., 2012*). The present study offers some insights in this regard. While the total score of the impulsivity scale was a strong predictor of goal-directed deficits, it did not load significantly onto the 'Compulsive Behavior and Intrusive Thought' factor, suggesting it has an independent association with goal-directed deficits. The supervised analysis identified the items from the impulsivity scale that best predicted goal-directed deficits. In terms of the overlap between the impulsivity questionnaire items and Factor 2, the two above-threshold predictors of model-based deficits were "I spend or charge more than I earn" and "I do things without thinking", each of which is qualitatively characteristic of compulsive, habitual behavior. Importantly, the three items that did *not* overlap with 'Compulsive Behavior and Intrusive Thought', but still predicted model-based learning, tracked subjects' general interest in engaging with the task (e.g. "I do not like puzzles", "I do not like to think about complex problems"). We suggest that these items may not be of particular clinical importance, but simply serve as a marker of how likely individuals are to engage with the task material. In summary, while a small subset of the impulsivity items contributed to 'Compulsive Behavior and Intrusive Thought', impulsivity as assessed by our scale was mostly distinct. Of course, impulsivity as a construct itself involves a broad range of potentially distinct behaviors, such as impatient inter-temporal choice preferences and premature responding (*Dalley et al., 2011*). Further work will be need to assess how such behaviors relate to the features measured here; notably, our large-scale online methodology is well suited for examining such questions.

As has been shown for other tests that broadly fall within the category of executive function (*Arffa, 2007*), model-based learning was also associated with IQ and age (and gender in experiment 2 only). Although these effects were controlled for in all analyses and therefore do not bias the interpretation of our results, they highlight the fact that the coupling between model-based learning and 'Compulsive Behavior and Intrusive Thought' is far from perfect. One particularly interesting observation is that as people get older, they show greater deficits in model-based learning (*Supplementary file 1B*), but fewer psychiatric symptoms on all nine questionnaires collected in the present study (*Supplementary file 1A*), in line with prior work with diagnosed patients (*Kessler et al., 2005*). This incongruence suggests that there may be multiple dissociable processes responsible for model-based learning. Future studies are needed to dissect this somewhat complex construct into its constituent parts (as has been already attempted for other executive tasks [*Miyake et al., 2000*]), with a view to identifying the simpler component that is specific to the compulsive phenotype. Relatedly, future work might test if working memory might conceivably contribute to this association observed in the present study (*Otto et al., 2013*). Also, the strength of the association between a clinical phenotype and an underlying mechanism is fundamentally limited by the accuracy with which we can assess that phenotype. Aside from issues of relatively low reliability of self-report clinical symptoms (e.g. self-report OCD, $r=0.71$ [*Hajcak et al., 2004*]), we are also limited by the questions we ask. For example, in the present study we did not account for pathological gambling or trichotillomania, which are similarly defined clinically by a loss of control over repetitive behavior (*Potenza, 2008*; *Chamberlain et al., 2007*) and therefore may contribute noise to our signal. It is clear that iterative improvements to both self-report assessment and behavioral testing are needed to increase effect sizes and further refine the neurobiological characterization of Compulsive Behavior and Intrusive Thought suggested by these data.

Although we have labeled the three factors that emerged from our unsupervised analysis based on theoretical considerations, we acknowledge that this is an inherently subjective process and that some may rightfully disagree with our choice of terminology. An important distinction to be made here is that although this labeling process was subjective, the way in which these clusters were *identified* was not. We first identified a heretofore-unrecognized collection of trans-diagnostic psychiatric symptoms based on their inter-correlations and then validated this clustering by demonstrating an association with neurocognitive performance in an independent task. 'Compulsive Behavior and Intrusive Thought' is not intended to be a fixed or final definition – rather it is hoped that future work can (i) use the clusters defined in this study to find closer links between biological markers and clinical and (ii) improve and augment these clusters through further data-driven evaluations. More broadly, we hope that this methodology can be employed in many other areas of psychiatry where the considerable issues of heterogeneity within and homogeneity across the existing diagnostic categories is curtailing efforts to delineate the precise neurobiological basis of psychiatric problems.

In the present study, we did not screen for psychiatric disorders, favoring the acquisition of a large sample within which we could leverage normal variation in psychopathology. Although our results converge with prior work using this neurocognitive marker in compulsive disorders (*Voon et al., 2015*), future studies will be needed to test if these dimensional results map onto clinically diagnosed patients. For example, based on the results of the present study, we hypothesize that the co-morbidity between OCD and addiction might be largely explained by a common deficit in goal-directed control. Conversely, the co-morbidity between OCD and anxiety disorders might be explained by an orthogonal (equally important) symptom dimension. This kind of exciting work should be coupled with studies aiming to use such trans-diagnostic markers to predict treatment response on an individual basis within the existing diagnostic categories.

Altogether, these data suggest that 'Compulsive Behavior and Intrusive Thought' together constitute a dimensional psychiatric phenotype that can be tracked in the general population and is linked to deficits in goal-directed control over action, which has a clear neurobiological foundation (*Dolan and Dayan, 2013*). These data highlight the utility of a computational approach to psychiatry (*Montague et al., 2012*) and specifically our novel approach of leveraging large datasets, online testing, and normal variation in psychopathology to isolate the neurocognitive basis of psychiatric dimensions that may be relevant for multiple disorders. More broadly, the results of this study constitute progress toward realizing the promise of the RDoC initiative, suggesting that dimensional markers of psychiatric disturbances may map more closely to underlying biological states than do the overlapping and heterogeneous definitions of DSM disorders.

## Materials and methods

### Participants

Data were collected online using Amazon's Mechanical Turk (AMT). Participants were paid a base rate (Experiment 1: $2, Experiment 2: $2.50) in addition to a bonus based on their earnings during the reinforcement-learning task (In each experiment, M=$0.54, SD=0.04). Subjects were based in the USA (i.e. had a US billing address with an associated US credit card, debit card or bank account), 95% of their previous tasks were approved and were 18 years or older. Participants in Experiment 1 (N=548) were 357 females (65%) and 191 males with ages ranging from 18 to 72 (M=35, SD=11). Using the effect size of the relationship between OCD symptoms and model-based learning observed in Experiment 1, we estimated that to achieve 80–90% power on a two-tailed test with a significance level of p<0.05, the sample size needed in Experiment 2 between 1223–1637 subjects. Experiment 2, participants (N=1413) were 823 females (58%) and 590 males with ages ranging from 18 to 76 (M=33, SD=11). The research team did not know participants' identities; participants provided their consent online by clicking 'I Agree' after reading the study information and consent language in accordance with procedures approved by the New York University Committee on Activates Involving Human Subjects.

### Exclusion criteria

In line with suggestions made in the literature with respect to studies conducted using Amazon's Mechanical Turk (AMT), several *a priori* exclusion criteria were applied to ensure data quality (*Crump et al., 2013*). Prior to completing the RL task subjects completed a practice phase, which consisted of written instructions, passively viewing 20 trials demonstrating the probabilistic nature of the associations between second stage fractals and subsequent 25c rewards, and actively participating in 20 trials demonstrating the probabilistic transition structure of the task (i.e. selecting a top-stage box on each trial and observing the transition to second-stage states). After this practice phase, participants were required to correctly answer a 3-item basic comprehension test regarding the rules of the reinforcement-learning task (*Gillan et al., 2015c*). If subjects failed to answer the questions correctly, they were sent back to the beginning and required to repeat the instructional section prior to re-taking the comprehension test. Participants were permitted to repeat this cycle as many times as was necessary for them to pass this test and continue to the main experiment.

The RL instructions and associated comprehension test were always administered first, followed by the RL task, then the IQ test and finally the self-report psychiatric assessments. Within the self-report section, the order of the questionnaires was fully randomized. Exclusions based on task performance/engagement were applied sequentially, in the order listed below. *Reinforcement-Learning Task Exclusion Criteria*: Subjects were excluded if they missed more than 10% of trials (Exp1: n=11; Exp2: n=62), responded on the same key on more than 95% of trials on which they registered a response (Exp1: n=46; Exp2: n=85) or had implausibly fast reaction times, i.e. ± 2 standard deviations from the mean (Exp1: n=9; Exp2: n=18). *Clinical Questionnaires Exclusion Criterion*: In an effort to identify participants who were not reading the questions prior to selecting their responses, we included one catch item: "If you are paying attention to these questions, please select 'A little' as your answer". Very few subjects failed to select the appropriate response to this catch question; those that did were excluded (Exp1: n=0; Exp2: n=6). *IQ Test Exclusion Criterion*: Participants who did not answer correctly to any of the IQ questions were excluded from further analysis (Exp1: n=32; Exp2: n=87). The adaptive character of the test meant that participants responding incorrectly received increasingly easy items; consistently failing to respond correctly indicates that given participants might have been inattentive or dishonest. In total, 98/646 (15%) subjects who submitted data were excluded in Experiment 1 and 258/1671 (15%) were excluded in Experiment 2.

We tested *post hoc* if subjects excluded on the basis of RL task performance were typical in terms of psychiatric self-report and other assessments. In study 1, we found that those subjects who were excluded had lower symptoms of OCD (t(604)=2.477, p=0.014), trait anxiety (t(604)=2.225, p=0.027), and a trend towards lower levels of depression (t(604)=1.799, p=0.073). These differences were not observed in Study 2, where all questionnaire total scores were not significantly different across groups (p>0.05). For both Experiment 1 and 2, results presented in this paper are not changed by the inclusion of these subjects in the analyses.

## Reinforcement learning task

To assess individual differences in goal-directed learning, we used a reinforcement-learning task (*Daw et al., 2011*) that distinguishes goal-directed ('model-based') learning from basic temporal difference ('model-free') learning. Model-based learning, like 'goal-directed learning', reflects the extent to which individuals integrate contingency information with estimations of outcome value to make choices, and predicts whether or not individuals can exert control over their habits in a devaluation test (*Gillan et al., 2015c*; *Friedel et al., 2014*). While model-free learning has been suggested to capture slow incremental learning characteristic of habit-formation itself, empirical studies using sequential decision tasks have not detected this relationship (*Gillan et al., 2015c*; *Friedel et al., 2014*), and this converges with the empirical observation that deficits in model-based (but not model-free) learning have been observed in compulsive disorders (*Voon et al., 2015*). The design of the task is presented in *Figure 1*. On each trial, subjects were presented with a choice between two fractals (2.5 s choice window). Each fractal usually (i.e. 'common' transitions: 70%, *Figure 1A*, white arrow) led to a particular second state (orange or blue) displaying another two fractal options. Selecting one of the fractals in the second stage resulted in participants being probabilistically rewarded with a picture of a 25¢ coin. There was a unique probability of receiving a reward associated with each second stage fractal, and these drifted slowly and independently over time (never being less than 0.25 or greater than 0.75). Responses were indicated using the left ('E') and right ('I') keys. Critically, on 30% of 'rare' trials (*Figure 1A*, grey arrow), choices uncharacteristically led to the alternative second state. A purely 'model-free' learner makes choices based solely on whether or not they were rewarded the last time they performed this action, regardless of whether the transition was rare or common (*Figure 1C*). A 'model-based' learner, in contrast, makes decisions based not only on the history of reward, but also the transition structure of the task, i.e. the environmental contingency (*Figure 1D*). For example, if a choice was followed by a rare transition to a second state, and that second state was rewarded, a model-based learner would be more likely to choose the alternate action on the next trial, because this is more likely to return them to that rewarding second state. A model-free learner, on the other hand, would be more likely to repeat that same action again, making no adjustment based on the transition type. We used a logistic regression based on this logic to identify from their switching patterns the extent to which each participant exhibited goal-directed (model-based, vs. model-free) choices (*Daw et al., 2011*).

## IQ - Progressive matrices

Intelligence Quotient (IQ) was approximated using a Computerized Adaptive (CAT) based on a bank of n=26 items similar to those used in Raven's Standard Progressive Matrices (SPM: [*Raven, 2000*]). The item bank was built using two parameter logistic Item Response Theory model (2pl: [*Baker, 1992*]). Item parameters were estimated using an online piloting sample of 760 participants (not included in the present study) that took both the test used in this study and original SPM. Items retained in the item bank were characterized by parameters (item-fit and discrimination) comparable or better than original SPM items. The length of the CAT test was 5 items (plus one non-diagnostic starting items). The items, including the starting item, were selected using Maximum Fisher Information criterion (va der Linden et al.) with a randomesque parameter of n=3 (*Kingsbury and Zara, 1989*). The scores were estimated using a Bayes Modal estimator (*Birnbaum, 1969*). Estimates based on the piloting sample showed that the score based on a 5-item CAT correlates relatively highly (r=0.77) with a score of a full SPM test.

## Self-report psychiatric questionnaires

In both Experiments 1 & 2, subjects completed self-report questionnaires assessing obsessive-compulsive disorder (OCD) using the Obsessive-Compulsive Inventory – Revised (OCI-R) (*Foa et al., 2002*), depression, using the Self-Rating Depression Scale (SDS) (*ZUNG, 1965*) and trait anxiety was assessed using the trait portion of the State-Trait Anxiety Inventory (STAI) (*Spielberger et al., 1983*). In Experiment 2, subjects were additionally assessed for alcohol addiction using the Alcohol Use Disorder Identification Test (AUDIT) (*Saunders et al., 1993*), apathy using the Apathy Evaluation Scale (AES) (*Marin et al., 1991*), eating disorders using the Eating Attitudes Test (EAT-26) (*Garner et al., 1982*), impulsivity using the Barratt Impulsivity Scale (BIS-10) (*Patton et al., 1995*), schizotypy scores using the Short Scales for Measuring Schizotypy (*Mason et al., 2005*) and social

anxiety using the Liebowitz Social Anxiety Scale (LSAS) (*Liebowitz, 1987*). Means of these total scores are presented in *Supplementary file 1A*, along with their relationship to age, gender and IQ. In Experiment 2, subjects also completed some additional self-report assessments that were unrelated to the present study and will be published elsewhere. These self-report assessments were fully randomized within the psychiatric assessment component of the procedure.

## Quantifying model-based learning (Logistic regression)

Logistic regression analyses were conducted using mixed-effects models implemented with the *lme4* package in the R programming language, version 3.1.1 (http://cran.us.r-project.org). The model tested if subjects' choice behavior (coded as switch: 0; stay: 1, relative to the previous choice) was influenced by Reward (coded as rewarded: 1; unrewarded: -1), Transition (coded as common: 1, rare: -1), and their interaction, on the preceding trial. A main effect of reward indicates that there is a significant contribution of model-free learning to choice behavior. An interaction between Reward and Transition indicates that there is a significant contribution of model-based learning to choice behavior. Within-subject factors (the intercept, main effects of reward and transition, and their interaction) were taken as random effects, i.e. allowed to vary across subjects. First, we tested our basic logistic regression model, which included age, gender and IQ as fixed effects covariates. We used Bound Optimization by Quadratic Approximation (bobyqa) with 1e5 functional evaluations. The model was specified in the syntax of R as follows:

Stay ~ Reward * Transition * (IQz + Agez + Gender) + (Reward * Transition + 1 | Subject)

In each Experiment, we found a significant main effect of Reward ('model-free') and a significant Reward x Transition interaction ('model-based') (*Figure 4*, *Supplementary file 1B*). There was also an unhypothesized significant main effect of Transition and an interaction between Transition and IQ, such that subjects were more likely to stay following a common transition and individuals higher in IQ showed this pattern more strongly. This seemingly anomalous effect is likely a side effect of additional structure in the choices that the regression model fails to capture. In particular, in the full computational model, choices are impacted by incremental learning that accrues over trials, such that a choice on some trial is affected by rewards on multiple preceding trials. Although the regression model considers only the most recent trial's rewards, some aspects of additional learning might be correlated with the transition term, producing small bias that can be detected given the large sample size of the current study (*Skatova et al., 2013*). For instance, the full model tends to encounter a negative reward prediction error immediately following a rare transition, which is driven by learning about second-stage state values driven by rewards received on previous trials. Such structure is more interpretably subsumed within the model-based and model-free learning terms in the fits of the fuller computational model, where, notably, the key results were all recapitulated (see below).

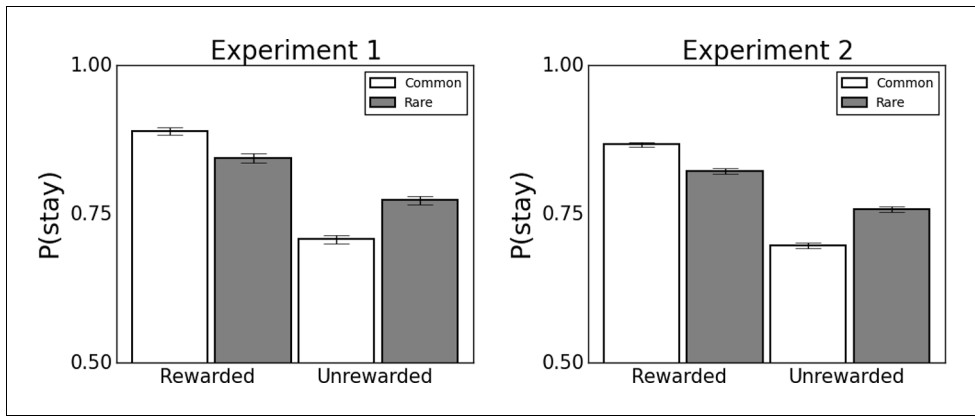

**Figure 4.** Behavioral data from experiments 1 (N=548) and 2 (N=1413). Error bars denote standard error. Data illustrate that consistent with previous studies (*Daw et al., 2011*), participants use a mixture of model-based and model-free learning to guide choice. Associated statistics are presented in *Supplementary file 1B*. *p<0.05 **p<0.01 ***p<0.001.

## Model-based learning and self-report clinical phenomenology

To test the hypothesis that a symptom severity of a given clinical construct ('SymptomScore') was associated with model-based learning deficits, we included the total score for each questionnaire (z-scored) as a between-subjects predictor and tested for interactions with all other factors in the model. We included age, gender and IQ (all z-scored) as fixed effects predictors interacted with Reward, Transition and Reward x Transition, to control for potentially confounding relationships between model-based learning and these covariates of no-interest. We hypothesized that there would be a significant three-way interaction between Reward, Transition and SymptomScore, only if those symptoms pertained to compulsive patterns of behavior. Specifically, we expected that greater severity of self-reported compulsive symptoms (i.e. OCD, addiction, eating disorders and aspects of impulsivity) would be predictive of reductions in model-based control over action. In the syntax of the *lme4* package, the specification for the regression was the same as above with the addition of the SymptomScorez, as follows:

Stay ~ Reward * Transition * (SymptomScorez + IQz + Agez + Gender) + (Reward * Transition + 1 | Subject)

In Experiment 1, three models were tested in which 'SymptomScorez' refers to the z-scored OCD, Trait Anxiety and Depression total scores in each respective model. Additionally, in Experiment 1, we also tested a model where self-report symptoms of OCD, trait anxiety and depression were included in the same model, to illustrate that the association with OCD symptoms survived the exclusion of shared variance. This was specified as follows:

Stay ~ Reward * Transition * (OCDz + TraitAnxietyz + Depressionz + IQz + Agez + Gender) + (Reward * Transition + 1 | Subject)

In Experiment 2, due to the high correlations across the different clinical scales, including all of the questionnaires in the same model would not produce an interpretable result - such that meaningful shared variance would be lost. Therefore, the associations between model-based learning and each questionnaire were assessed using separate models for each questionnaire (SymptomScorez, as specified above). As expected based on prior literature in this area (*Voon et al., 2015*), there was no relationship between clinical symptomatology and model-free learning in either Experiment (*Supplementary file 1C*). Note that we tested this model without gender (as gender was not itself significant in the model), and the results do not change - the effect of OCD symptoms on model-based learning remains significant ($\beta=-0.041$, SE=0.02, p=0.043). We nonetheless include gender in the presented models for Experiment 1 for consistency with Experiment 2, where gender effects were observed.

## Factor analysis

In order to (i) reduce the collinearity between the total scores for each of the 9 questionnaires employed and (ii) investigate the possibility that a more parsimonious latent trans-diagnostic structure could explain item-level responses in this dataset, we employed factor analysis using Maximum Likelihood Estimation (MLE). Factor analysis was conducted using the factanal() function from the Psych package in R, with an oblique rotation (oblimin). Two hundred and nine individual questionnaire items were entered as measured variables into the factor analysis. As responses on the schizotypy scale were binary at the item-level, a heterogeneous correlation matrix was computed using the hector function in polycor package in R. This allowed for Pearson correlations between numeric variables, polyserial correlations between numeric and binary items and polychoric correlations between binary variables. Factor selection was based on Cattell's criterion (*Cattell, 1966*); wherein a sharp transition from horizontal to vertical ('elbow") indicates that there is little benefit to retaining additional factors. The scree-plot was analyzed using an objective implementation of this criterion, the Cattell-Nelson-Gorsuch (CNG) test, which computes the slopes of all possible sets of three adjacent eigenvalues and determines the point at which there is the greatest differences in slope (nFactors package in R) (*Gorsuch and Nelson, 1981*). The CNG test indicated the existence of a 3-factor latent structure (*Figure 5*), which comprises factors that we labeled 'Anxious-Depression', 'Compulsive Behavior and Intrusive Thought' and 'Social Withdrawal' based on the strongest individual item loadings (*Supplementary files 2A, 2B & 2C*, respectively). Although Cattell's criterion is perhaps the most widely utilized rule-of-thumb for factor selection, we acknowledge that there are many alternatives and indeed another objective method, 'Parallel Analysis' (*Drasgow and Lissak, 1983*),

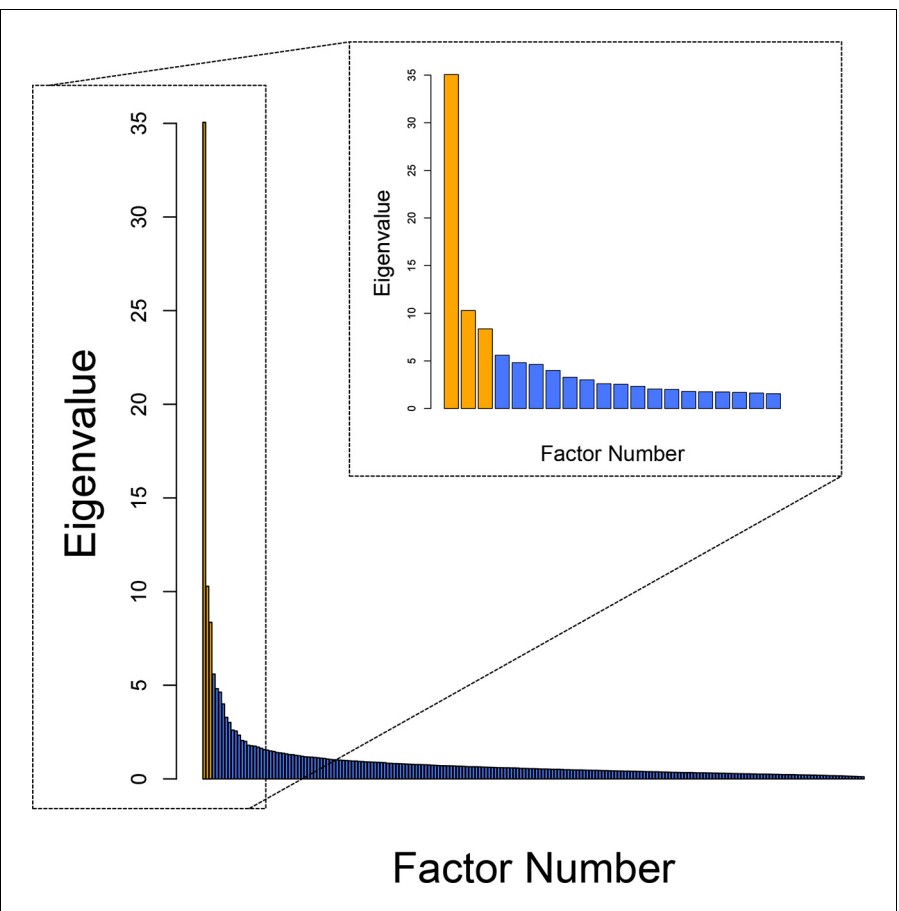

**Figure 5.** Scree plot of eigenvalues. The outer frame shows the eigenvalues for every possible factor solution, N=209. Inset is data for the first 20 potential factor solutions only. An empirically defined elbow, where Eigenvalues begin to level out, was identified at factor 4 using the nFactors package in R, provideing evidence for a 3-factor solution (*Cattell, 1966*), indicated in orange.*p<0.05 **p<0.01 ***p<0.001.

suggests an 8-factor solution to our data. This model was not only less parsimonious than the 3-factor solution, but in addition, a *post hoc* analysis revealed that it was also quantitatively inferior at predicting task performance when these 8 factors were entered as predictors in a mixed effects model (as per our main task analyses). Specifically, both Akaike Information Criterion (AIC) and Bayesian Information Criterion (BIC) were lower for the mixed effects model with covariates derived from the 3-factor solution relative to the 8-factor solution, indicating that this model was the best at predicting behavior.

## Labeling factors 1 and 3

As outlined in the results section, factors were labeled based on items that loaded most strongly and consistently. For the 'Anxious-Depression' factor, the highest average loadings came from the Trait Anxiety questionnaire (M=0.52, SD=0.17), followed by Apathy (M=0.44, SD=0.16) and Depression (M=0.38, SD=0.23) (*Table 2*). No other questionnaires reached the 0.25 average loading threshold we apply throughout this manuscript, but impulsivity came very close (M= 0.24, SD=0.22). Those impulsivity items that loaded most consistently reflected a tendency to not plan for the future and reduced ability to concentrate.

Factor 3 was labeled 'Social Withdrawal'. This factor was dominated by items from the Social Anxiety questionnaire (M=0.57, SD=0.14), and interestingly did not have a significant contribution from trait anxiety (M=0.13, SD=0.17). We chose the term 'withdrawal' primarily to distinguish this factor from the original social anxiety disorder questionnaire. Interestingly this factor had borderline

negative contributions from the alcohol addiction scale, which were low but consistent (M=−0.23, SD=0.06). Overall, this factor describes a phenotype that fears and avoids social situations, but interestingly also thinks excessively about future events and appears risk averse.

## Dimensional factors predicting model-based learning

A mixed effects logistic regression analysis was conducted to test the extent to which 'Anxious-Depression', 'Compulsive Behavior and Intrusive Thought', and 'Social Withdrawal' factors predicted deficits in goal-directed control over action. Specifically, these three factors were entered as z-scored fixed effect predictors in the basic model described above (i.e. interacted with Reward, Transition and Reward*Transition), while controlling for age, gender and IQ:

Stay ~ Reward * Transition * (Factorz + IQz + Agez + Gender) + (Reward * Transition + 1 | Subject)

The extent to which a factor is related to deficits in goal-directed control is indicated by the presence of a significant Reward*Transition*Factorz interaction (in the negative direction). Unlike the analysis of the original questionnaire total scores, in addition to testing the predictors separately in independent models, here we also tested a model where all three clinical predictors were included in the same model, which allowed us to statistically compare their effect sizes and thereby make claims about the specificity of our effects to compulsive (versus non-compulsive) aspects of psychopathology. *Table 3* shows the effects for model-based learning. There were no effects on model-free reinforcement learning.

To test the extent to which these results reflect a continuous relationship between model-based learning and Factor 2 ('Compulsive Behavior and Intrusive Thought'), constructed subsets of our total sample comprising either 'putative patients' (defined as those who scored in the top 25% on a given self-report measure) or subjects in the normal range (bottom 75%). We then repeated our analyses in these sub-samples. The slopes of the regression lines were consistent across all analyses, such that the relationship between model-based deficits and Factor 2 were observed both in individuals reporting the most severe symptoms and those in the normal range (see *Supplementary file 3*). In all 9 analyses with individuals in the 'normal range', the relationship between Factor 2 and model-based deficits were significant at p<0.05. In 5/9 analyses with 'probably patients' who were in the top 25% of symptom severity, the relationship between Factor 2 and model-based deficits were significant at p<0.05. This analysis had just ¼ of the total sample and was therefore severely underpowered. But nonetheless, the direction and slope of the effect were consistent across the board, providing evidence to suggest that these relationships will likely generalize to patient populations.

## Supervised analysis

In addition to the factor analysis, we also carried out a fully supervised analysis to identify the individual items that explained the most independent variance in goal-directed learning using linear regression with elastic net regularization. Elastic Net (*Zou and Hastie, 2005*) regularization imposes a hybrid of both $L_1$- and $L_2$-norm penalties (i.e., penalties on the absolute ($L_1$ norm) and squared values of the β weights ($L_2$ norm)). This allows relevant but correlated coefficients to coexist in a sparse model fit, by doing automatic variable selection and continuous shrinkage simultaneously, and selects or rejects groups of correlated variables. Least absolute shrinkage and selection operator LASSO, (*Tibshirani, 1996*) and ridge regression (*Hoerl and Kennard, 1970*) are special cases of the Elastic Net. The dependent measure in this analysis was each subject's model-based score (i.e individual subject's coefficients for reward x transition, corrected for age, IQ and gender, from the analysis in Experiment 2, *Supplementary file 1B*). All predictor data were first feature scaled (z-score transformed). We implemented ten-fold cross-validation with nested cross-validation for tuning and validating the model. Briefly, to implement cross-validation, the data were randomly split into 10 groups. A model was then generated based on 9 training groups, and then applied to the remaining independent testing group. Each group served as the testing group once, resulting in 10 different models, and predictions for every subject based on independent data. Nested cross-validation involved subdividing the 9 training groups (i.e., 90% of the sample) into a further 10 groups ('inner' folds). Within these 10 inner folds, 9 were utilized for training a model over a range of 50 alpha (0.01–1) and 50 lambda (0.0001–1) values, where alpha is the weight of lasso versus ridge optimization and lambda is the regularization coefficient. This generated a resulting model fit on the inner

fold test set for each possible combination of alpha and lambda. The mean fit over all 10 inner folds for each combination of alpha and lambda was then calculated and then used to determine the optimal parameters for the outer fold. We conducted 100 iterations of regularization with tenfold validation and retained items that were significant predictors of model-based learning in >=95% of final models. The overall model was significant, with the median cross-validated p=0.00003, median cross-validated *r*=0.11. Twenty-eight features met these criteria and are listed in *Supplementary file 4*.

## Quantifying model-based learning (Computational model)

The logistic regression analyses presented are a simplified method for analyzing the data, but as this approach only considers events taking place on the trial immediately preceding choice, it does not fully capture the influence of slow, incremental learning that takes place over many trials. These analyses have been shown to produce very similar results, particularly when estimating model-based learning (*Gillan et al., 2015c*; *Otto et al., 2013*) (indeed they are correlated at 0.87 here). Nonetheless, to complement these analyses, we verified that the relationship between model-based learning and compulsive behavior holds in the full computational instantiation of model-based and model-free reinforcement learning. For this analysis, choices were modeled as arising due to the weighted combination of model-free and model-based reinforcement learning. The model is equivalent to that used by Otto *et al* (*Otto et al., 2013*), which is itself a simplified variant of the one used by Daw *et al* (*Daw et al., 2011*). At each trial $t$, a participant makes a stage-1 choice $c_{1,t}$, occasioning a transition to a stage-2 state $s_t$ where she makes another choice $c_{2,t}$ and receives reward $r_t$. At stage 2, subjects are assumed to learn a value function over states and choices, $Q_t^{stage2}(s, c)$, whose value for the chosen action is updated in light of the reward received at each trial according to a delta rule, $Q_{t+1}^{stage2}(s_t, c_{2,t}) = (1 - \alpha)Q_t^{stage2}(s_t, c_{2,t}) + r_t$. Here, $\alpha$ is free learning rate parameter, and (in this and all analogous update equations throughout) we have omitted a factor of $\alpha$ from the last term of the update, which is equivalent to rescaling the rewards and $Q$s by $1/\alpha$ and the corresponding weighting parameters $\beta$ by $\alpha$. (*Otto et al., 2013*) The probability that a subject will make a particular stage-2 choice is modeled as governed by these choices according to a logistic softmax, with free inverse temperature parameter $\beta^{stage2}$: $P(c_{2,t} = c) \, \alpha \, \exp\left(\beta^{stage2} Q_t^{stage2}(s_t, c)\right)$, normalized over both options $c$.

Stage-1 choices are modeled as determined by the weighted combination of both model-free and model-based value predictions about the ultimate, stage-2 value of each stage-1 choice. Model-based values $Q^{MB}$ are given by the learned values of the corresponding stage-2 state, maximized over the two actions: $Q_t^{MB}(c) = \max_{c2}\left(Q_t^{stage2}(s, c_2)\right)$, where $s$ is the stage-2 state predominantly produced by stage-1 choice $c$. Model-free values are learned by two learning rules, each of which updates according to a delta rule with a different estimate of the second-stage-value: TD(0), where $Q_{t+1}^{MF0}(c_1, t) = (1 - \alpha)Q_t^{MF0} + Q_{t+1}^{stage2}(s_t, c_{2,t})$, and TD(1), where $Q_{t+1}^{MF1}(c_1, t) = (1 - \alpha)Q_t^{MF1} + r_t$. Stage-1 choice probabilities are then given by a logistic softmax, with contributions from each value estimate, each weighted by its own free inverse temperature parameter: $P(c_{1,t} = c) \propto \exp\left(\beta^{MB}Q_t^{MB}(c) + \beta^{MF0}Q_t^{MF0}(c) + \beta^{MF1}Q_t^{MF1}(c) + \beta^{stick}I(c = c_{1,t-1})\right)$. Here, $I(c = c_{1,t-1})$ is a binary indicator for the choice that repeats the one made on the previous trial; the corresponding weight $\beta^{stick}$ measures the tendency to alternate or perseverate regardless of feedback.

At the conclusion of each trial, the value estimates $Q$ (of all three sorts) for all unchosen actions and unvisited states are decayed multiplicatively by $(1 - \alpha)$.

Altogether, the model has six free parameters: five weights $\beta$ and a learning rate $\alpha$. These represent a minor change of variables with respect to the equations in *Otto et al. (2013)*: In particular, by separating the TD(0) and TD(1) stages of the model-free update into separate $Q$ values, we split Otto et al.'s aggregate model-free weight $\beta^{MF}$ version into two variables, thereby also replacing their eligibility trace parameter $\lambda$ which encodes the balance between the two updates and eliminating the (0,1) boundaries associated with that variable. Following estimation, we reconstruct the aggregate model-free weighting as $\beta^{MF} = \beta^{MF0} + \alpha\beta^{MF1}$, where the factor of $\alpha$ accounts for the difference in scaling between the two weights arising from the omission of $\alpha$ from the update equations.

For each participant, we estimated the free parameters of the model by maximizing the likelihood of her sequence of choices, jointly with group-level distributions over the entire population using an

Expectation Maximization procedure (*Huys et al., 2011*) implemented in the Julia language (*Bezanson et al., 2012*). We extracted the per-subject model-based and model-free weightings $\beta^{MB}$ and $\beta^{MF}$ as indices of the strength of each sort of learning for further analysis of individual differences. Specifically, we used subject-level estimates of model-based and model-free learning from the computational model as dependent variables in regression analyses where clinical characteristics (i.e. questionnaire total scores and factors from factor analysis) were independent variables. The results of the full reinforcement-learning model mirrored that of the logistic regression analysis in almost every respect. The two differences were that when estimated using the computational model, the relationship between self-report OCD symptoms and goal-directed learning in Experiment 1 fell short of reaching significance at $p < 0.05$ (*Supplementary file 5A*). The size of this effect was similar in Experiment 2, but with the benefit of an increased sample size was highly significant, indicating this was an issue of statistical power. Secondly, while Schizotypy did not reach significance as a predictor of model-based deficits using the regression model, it was a significant predictor model-based learning when computationally estimated. There were no relationships between self-report psychopathology and model-free learning defined using the computational model. A side-by-side comparison of the predictive power of model-based learning defined using the computational model versus one-trial back regression analysis is presented in *Supplementary file 5B*.

## Acknowledgements

Funded by a Sir Henry Wellcome Postdoctoral Fellowship (101521/Z/12/Z) awarded to CM Gillan.

## Additional information

### Funding

| Funder | Grant reference number | Author |
|---|---|---|
| Wellcome Trust | 101521/Z/12/Z | Claire M Gillan |
| National Institute on Drug Abuse | 1R01DA038891 | Nathaniel D Daw |
| James S. McDonnell Foundation | Scholar Award | Nathaniel D Daw |

The funders had no role in study design, data collection and interpretation, or the decision to submit the work for publication.

### Author contributions

CMG, Conception and design, Acquisition of data, Analysis and interpretation of data, Drafting or revising the article; MK, Acquisition of data, Drafting or revising the article; RW, Analysis and interpretation of data, Drafting or revising the article; EAP, Conception and design, Drafting or revising the article; NDD, Conception and design, Analysis and interpretation of data, Drafting or revising the article

### Author ORCIDs

Claire M Gillan, http://orcid.org/0000-0001-9065-403X

### Ethics

Human subjects: Participants provided their consent online after reading the study information and consent language in accordance with procedures approved by the New York University Committee on Activates Involving Human Subjects.

# Additional files

**Supplementary files**

• Supplementary file 1. (A) Questionnaire Total Scores in Experiments 1 and 2. [a] For the purposes of comparison across studies using the EAT, responses on the 6-point scale are converted as follows: (1:0, 2:0, 3:0, 4:1, 5:2, 6:3) and totaled to produce the mean reported above (Everitt and Robbins, 2005). For analysis purposes (including correlations reported above), however, the continuous (i.e. 1,2,3,4,5,6) values were used (M=33.31, SD=17.36). Likewise, for LSAS, the mean reported above corresponds to the summed total of the full questionnaire - i.e. including answers to both avoidance and fear/anxiety probes. However, for the purposes of analyses, we used the average of the avoidance and fear/anxiety answers for each item. [b] Positive t-values indicate higher scores among males, while negative t-values indicate higher scores in females. (B) Results from Basic Logistic Regression Model in Experiment 1 (N=548) and Experiment 2 (N=1413) with Age, Gender and IQ as fixed effects predictors. *p<0.05 ** p<0.01 ***p<0.001SE=standard error. (C) Questionnaire Total Scores and Model-Free Learning*p<0.05 ** p<0.01 ***p<0.001SE=standard error. Each row reflects the results from an independent analysis where each questionnaire total score (z-transformed) was entered as SymptomScorez in the following model: glmer(Stay ~ Reward * Transition * SymptomScorez + Reward * Transition * (IQz + Agez + Gender) + (Reward * Transition + 1 | Subject)). Model-free statistics refer to the following interaction: SymptomScorez x Reward. For each, positive β values indicate that the symptom score is associated with greater model-free learning, while negative β values indicate that the symptom score is associated with reduced model-free learning.

• Supplementary file 2. (A) 'Top loading items on 'Anxious-Depression' factor Summary of item loadings onto 'Anxious-Depression' factor. The top loading items from each questionnaire are displayed in descending order, provided they are above a threshold loading of +/- 0.25. Words in parentheses, e.g. "(do not)" are added here (but were not presented to participants) to facilitate interpretation of the direction of effects for items that are reverse-coded. (B) 'Top loading items on 'Compulsive Behavior and Intrusive Thought'. Summary of item loadings onto 'Compulsive Behavior and Intrusive Thought' factor. The top loading items from each questionnaire are displayed in descending order, provided they are above a threshold loading of +/- 0.25. (C) 'Top loading items on 'Social Withdrawal' factorSummary of item loadings onto 'Social Withdrawal' factor. The top loading items from each questionnaire are displayed in descending order, provided they are above a threshold loading of +/- 0.25. Words in parentheses, e.g. '(do not)' are added here (but were not presented to participants) to facilitate interpretation of the direction of effects for items that are reverse-coded.

• Supplementary file 3. The relationship between model-based learning and Factor 2 is broadly consistent across 'putative patients' (top 25%) and subjects scoring in the normal range (bottom 75%). Plotted here are regression lines indicating the strength of the relationship between model-based deficits and Factor 2 ('Compulsive Behavior and Intrusive Thought') in putative patients (top 25%, blue) and in subjects in the normal range (bottom 75%, red). Each subplot reflects a different subset of the population, based on that clinical questionnaire. Eighteen independent analyses were subsequently carried out (i.e. two per subplot).

• Supplementary file 4. Significant Predictors of Goal-Directed (Model-Based) Learning from Supervised Analysis Features that are significantly associated with model-based learning identified using elastic net regularization with tenfold cross-validation, observed in >95% of 100 iterations tested. Index refers to the item number from the questionnaire of origin. Beta refers to the coefficient from the regularized regression model. Words in parentheses, e.g. '(do not)' are added here (but were not presented to participants) to facilitate interpretation of the direction of effects for items that are reverse coded. The last column 'FA Loadings' indicates the significant overlap in terms of loading on Factors F1 ('Anxious-Depression'), F2 ('Compulsive Behavior and Intrusive Thought') or F3 ('Social Withdrawal'), in the positive (+) or negative (-) direction using a cut-off at loadings >= 0.25.

• Supplementary file 5. (A) Association between questionnaire total scores and model-based learning defined using full computational model. *p<0.05 ** p<0.01 ***p<0.001. Each row reflects the results

from an independent analysis where each questionnaire total score (z-transformed) was entered as SymptomScorez in the following model: lm(Model-Based-Learning ~ IQz + Agez + Gender + SymptomScorez). Statistics refer to the main effect of SymptomScorez on Model-Based-Learning, i.e. the extent to which that questionnaire total score is associated with changes in model-based learning (which was defined for each participant using the full computational model). (B) Comparing the predictive power of model-based learning defined using the computational model versus one-trial back regression analysis. Each row reflects the results from an independent analysis where each questionnaire total score (z-transformed) was entered as SymptomScorez in the following model: lm(SymptomScorez ~ ModelBasedScore). Prior to conducting these analyses, we regressed out the effects of age, gender and IQ so that we could directly compare the $r^2$ of the models. ModelBasedScore was derived from the one-trial back regression (first three columns) or the computational model (last three columns results). For each, positive β values indicate that the ModelBasedScore is associated with fewer symptoms, whereas negative β values indicate that the symptom score is associated with increased symptoms. The difference between the two approaches is negligible. However the computational model did produce nominally higher $r^2$ and lower $p$-values for the relationship between clinical scores and model-based learning.

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
