## [Decision Letter]

Thank you for submitting your work entitled "Compulsivity is a trans-diagnostic trait characterized by deficits in goal-directed control" for consideration by *eLife*. Your article has been reviewed by two peer reviewers, and the evaluation has been overseen by Reviewing Editor Michael Frank and Timothy Behrens as the Senior Editor. One of the two reviewers has agreed to reveal his identity:

Klaas Stephan.

The reviewers have discussed the reviews with one another and the Reviewing editor has drafted this decision to help you prepare a revised submission

Summary:

The authors conduct a comprehensive analysis from a large sample of subjects using questionnaire-based measures of clinical variables and two independent experiments using a sophisticated reinforcement learning task that dissociates goal-directed behavior from habitual stimulus-response learning. They report consistent demographic and clinical factors that underlie reductions in goal-directed behavior during learning. Supervised and unsupervised analysis of the link from questionnaire data to task performance points to compulsivity (and its various manifestations) as one key clinical factor that is related to reduced goal-directed behaviors.

Essential revisions:

Overall, the reviewers were impressed with the sophistication of the analysis and agreed that this study represents an important step toward large scale quantitative assessment of relevant phenotypes informed by computational cognitive neuroscience – that is, one of the main goals of computational psychiatry. They also agreed that the approach is original, exploits a large online sample (with appropriate controls for data quality), and is based on a systematic body of work, conducted by the authors over several years. However, they all expressed concerns with respect to the main take-home message of the manuscript that compulsivity is a trans-diagnostic factor that relates to deficits in model-based learning. This was particularly concerning given that other demographic factors (e.g., age) had even greater impact on the same measure of model-basedness, limiting the conclusions (and application) that one could garner from using this theoretically grounded construct clinically. Nevertheless, we all agree that the analysis is itself useful and sophisticated and we would like to see a revised manuscript that substantially tones down the main claim in the thrust of the motivation (including the title). We would be happy to consider a more nuanced, balanced and thorough (perhaps longer) characterization of the phenotype that is not as centrally focused on compulsivity, but rather presents a large scale analysis of factors that relate to MB vs. MF performance (and indeed other task measures not obviously MB per se, like the transition effect), including age, gender, IQ but also other clinical factors (impulsivity vs. compulsivity), whether these are independent or interactive factors, with discussions on whether they are likely to have similar or different mechanisms, etc. While this may seem like a major undertaking, we think that ultimately this kind of description can be more useful for showcasing the strength of your combination of theory-driven and data-driven approaches. Should you wish to maintain the stronger claim and message and focus on compulsivity we would suggest submitting elsewhere. Below these points are elaborated by comments from the individual reviewers, compiled together.

1) I've spent a substantial amount of time mulling over this paper, which really does have many strengths and is exemplary in many ways: a well-motivated task; applied to a large population; combined with an interesting methodology that makes an important step forwards in terms of relating neurobiological/cognitive mechanisms to psychopathology. The results are initially very intriguing – particularly those from the elastic net where impairments in goal-directed control seem to pick out symptoms of compulsivity and intrusive thoughts. However, on reading it more closely there are some important drawbacks which I think require the conclusions to be very significantly toned down; or additional analyses to substantiate them. This, in turn, might make it, in my view, more appropriate for less general journal. My major concerns are:

Age has much more of an effect than compulsivity – but OCD prevalence does not increase with age (e.g. Kessler et al. 2005, Arch Gen Psych 62(5):593-602). How can this be if goal-directed impairments underlie compulsivity in such a specific manner? The answer presumably is that goal-directedness depends on multiple processes, and that those related to compulsivity and age might in some way be dissociable. But does that not then make compulsivity a less specific guide to the underlying neurobiology? Isn't this also suggested by the fact that the relationship is, overall, quite weak: in the elastic net, the cross-validated correlation is 0.11? The temporal evolution of OCD decreasing with age also jars with the influence of age.

OCD, addiction, etc. are characterised by the positive presence of certain behaviors that bear the hallmark of 'habits'. Why does this not show up in the task? The possibility that the task seems to be insensitive to habitual variation (it never seems to show up in correlations despite the model-free prediction error regressors showing the strongest correlations with BOLD, i.e. neurobiology) somewhat questions the strong conclusions about compulsivity being specifically due to an impairment in goal-directed control: subjects could also have an impairment (excess?) in habitual learning (as one might conclude from excessive habitual behavior in e.g. Gillan et al., 2015 AJP), but this doesn't show up in the task because it's not sensitive. This again makes the conclusions they are drawing from the results just too strong. They state that the literature shows that deficits in model-based but not model-free decision-making has been found and cite Voon et al., Mol. Psych. 2014, but that study used the same task, hence not really addressing this point.

They make statements about patient populations but include neither patients nor any other measure by which functional impairment could be judged, and refer to diagnostic categories ('OCD', 'Alcohol addiction') despite not performing any diagnostic tests. The results, figures and discussion needs to avoid reference to diagnostic categories, and I find the term 'trans-diagnostic' difficult in the absence of any diagnosis. At the core if this is that it is unclear whether the results are driven by what one might observe in a typical patient population. One way to address this is to recapitulate the results only amongst those subjects with scores above cutoffs in any one measure, and then talk about 'putative patients' or so. We also need to know whether those subjects excluded based on performance were typical in terms of self-report.

There is no information about the stability of the effects over time, and hence the term trait is confusing. In fact, the covariates are mostly measures of state, not trait.

2) On closer inspection, the elastic net analysis is far less convincing than on reading the results – the strongest loadings (I tried to sort them in descending order from Table 3):

I feel that there are good and bad numbers;

Am preoccupied with the thought of having fat on my body;

I vomit after I have eaten;

I check things more often than necessary;

Am terrified about being overweight;

Like my stomach to be empty;

My heart beats faster than usual.

With overall only two items from the OCI-R (the measure of OCD used), and neither of these is being significantly loaded onto by the compulsivity factor. The fact that so many eating disorder items show up certainly deserves some comment beyond it being just another compulsive phenotype, but overall this just doesn't quite capture 'compulsivity'. Only one out of the top 8 items has anything obvious to do with compulsivity (other than referring to a disease which they labelled as compulsive).

3) I do wonder about how overall severity contributes. This is important because severity is strongly related to comorbidity (see e.g. Kessler et al., 2005, in the same volume as above), and hence important for any trans-diagnostic processes. Half the questionnaires are correlated (and picked up by the compulsivity factor). The most severely ill patients might thus be most likely to respond positively on many compulsivity items. Could it be that the most severely impaired patients simply look compulsive because they are more likely to have more comorbid disorders and hence show up in the compulsive category?

4) In the FA, the first component doesn't contain anxiety at all. Anxiety loads much more on the second factor, and does so possibly even more than compulsivity: there are around 9 or 10 items that clearly relate to anxiety loading onto it, but only 2 items relating to compulsive behaviours. A number of the AUDIT variables are hard to relate to compulsions: alcoholics start drinking early as they experience withdrawal symptoms after a night of sleep. If anything, this component is more related to obsessions, anxious worries and difficulties controlling thoughts – which is, in terms of constructs, much closer to goal-directed deficits, it seems to me.

5) The task itself isn't obviously specific as it is not clear what the model-free component quite captures. This makes it more of a shame they didn't test components we know impact on m-b choices, such as working memory or stress. Impairments in this are also 'trans-diagnostic', and it would have been nice to show that they don't have the specificity of g-d choices.

6) Both reviewers expressed concerns about the explanatory power (of excessive habit formation due to deficient model-based control) for understanding clinical aspects of compulsivity. As you outlined in the Introduction, a key motivation for studying the relation between model-based /goal-directed decision-making and compulsive symptoms is the notion that "a deficit in deliberative, goal-directed control may leave individuals vulnerable to rely excessively on forming more rigid habits". I understand why this is a straightforward and attractive perspective to explain certain aspects of compulsivity. However, I think it would also be appropriate to mention challenges and potential limitations of this perspective in the Discussion – particularly because the dimensional approach chosen here suggests applicability of the proposed mechanism to clinical phenomena. For example, how exactly would a putative deficit in model-based control lead to prominent symptoms in OCD, such as excessive checking, fear of germs, or desire for order? The nomenclature and with it the framing need quite some work, e.g. categorical/dimensional measures, in terms of state/trait distinction, and distinctions between compulsions and obsessions.

7) The paper is very well written and of beautiful simplicity – a pleasure to read. However, sometimes a few more technical details or conceptual distinctions may have to be included in the main text to avoid confusion. First, the Introduction repeatedly refers to unspecified "OCD symptoms" which I found confusing, given that the paper is about the general population and that numerous symptoms of OCD exist. I would recommend avoiding the clinical label OCD and referring to compulsivity instead, stating the specific questionnaire you used. Similarly, in the Results section (second paragraph), there is a tension between using trait labels (impulsivity, compulsivity) and diagnostic labels (eating disorders, alcohol addiction); the latter is confusing (and not quite appropriate), given that your study examines the general population. You could eliminate this tension and, at the same time, increase clarity by always referring to the scores of the respective questionnaires. Second, the Results section should define the measure of model-based learning used (first paragraph). Until I went through the Methods section, I was not sure how exactly model-based learning was operationalised, and whether you were referring to a behavioural readout or to the parameter estimates of a computational model.

8) You report analyses based on behavioural readouts (trial-by-trial stay/switch behaviour), not model parameter estimates, because the qualitative conclusions drawn from both types of analyses seemed to be almost equivalent. Does this also hold with regard to how well questionnaire scores can be predicted, or does the computational model have a competitive advantage there? It would be instructive for the technically interested reader if you could include estimates of predictive accuracy for both approaches, perhaps in the supplementary material.

9) In the subsection “Quantifying Model-based Learning (Logistic Regression)”, second paragraph: The significant main effect of Transition is very interesting. Could you please state the direction of this effect and perhaps even offer a (speculative) interpretation? This is another place in which a more thorough analysis of the factors on both sides (task measures and demographic/clinical variables) can be useful.

---

## [Author Response]

*Essential revisions: Overall, the reviewers were impressed with the sophistication of the analysis and agreed that this study represents an important step toward large scale quantitative assessment of relevant phenotypes informed by computational cognitive neuroscience* –

*that is, one of the main goals of computational psychiatry. […] Below these points are elaborated by comments from the individual reviewers, compiled together.*

We thank the reviewers and Reviewing Editor for their comments. We have made substantial changes to the manuscript in light of the issues raised by the reviewers. The main changes are summarized here:

1) We have changed the title, Abstract and body, making clear that the symptom dimension related to failures in goal-directed control is characterized by both compulsive behavior and associated repetitive thoughts (i.e. not just behavior), and more generally making clear that our goal is to delineate the scope of the psychiatric phenotype associated with failures of goal directed control, including or beyond compulsivity. Related to this issue, we speculate in more detail about the functional relationship between compulsions and obsessions in the Discussion, drawing on recent empirical work (Cushman and Morris, 2015 and Gillan et al., 2014).

2) We have made it more explicit throughout the manuscript that we are not studying diagnosed patients, but rather normal variation in self-report symptomatology. We do this by clearly referring to ‘self-report questionnaires’ at all points where we evoke the name a clinical diagnostic category such as ‘OCD’. Similarly, we make clear that we do not have any evidence for state vs. trait-dependence of these symptom dimensions. We have eliminated the word ‘trait’ from the manuscript (except when describing the ‘trait anxiety’ questionnaire). We also omit the term “trans-diagnostic.”

3) We have moved the age, IQ and gender findings to the main Results section and talk about these in more detail in the Discussion, thereby providing a more comprehensive analysis of what demographic (as well as psychiatric) features relate to task performance.

4) We have responded to all analytical queries, providing supplementary summary data, figures and statistics where appropriate.

We think that these changes deliver on the suggestion of a more balanced and broader exploration of the phenotype. That said, we do retain some of the framing and emphasis (though toned down) on the notion of compulsivity and its relationship to goal-directed control as the initial idea that launches our study.

This is for several reasons:

1) Several of the reviewers’ concerns with respect to how appropriate it was to label Factor 2 ‘compulsivity’ arose from problems with how we presented the results tables, which we believe led reviewers to underappreciate the strength of the evidence. Specifically, in the prior submission we displayed just the top 6 items from each questionnaire that loaded on a given factor. This evidently led one reviewer to believe certain items from the OCD scale did not load at all on Factor 2, when in fact they were amongst the highest loading items of all items (just not in the top 6). We have changed our presentation approach to avoid this misunderstanding and believe this resolves several of the reviewers’ concerns. There was a similar problem with the table presenting the results of the regularized regression, which we have corrected.

2) That being said, we agree with the reviewers that the label ‘compulsivity’ was not apt to describe Factor 2, because thought processes also contributed significantly to this factor. This should rightly be seen as a novel, positive finding of the study. Not directly mentioning these in the label, may serve to inappropriately diminish their contribution and this was not our intention. We have renamed the factor ‘Compulsive Behavior and Thought’, to reflect the tight coupling between compulsive behaviors and cognitive processes (i.e. interpretations, obsessions, preoccupations, unusual beliefs). We struggled somewhat with the nomenclature, which (as we say in the Discussion) is obviously subjective but also necessary simply for communication. If the appearance of the term “compulsive” in this more expansive and tentative label remains unsatisfactory we are happy to take further advice.

3) The design choices for our study (including things like the choice of questionnaires and the power analysis/sample size) are all driven by, and really only sensible given our specific motivating hypothesis. That hypothesis, while admittedly imperfect, is based on the intersection of: (i) a consistent body of our own work published over the past 5 years and (ii) recently highlighted issues with our current psychiatric taxonomy.

4) Relatedly, while we give more attention in both Results and Discussion to the additional findings regarding age and IQ, we think that nevertheless retaining greater emphasis to the psychiatric features is appropriate given that these are the novel findings. Age did not have a greater impact on model-based learning than Factor 2 (relevant data detailed below): age and Factor 2 were statistically equivalent. While we can appreciate that age and IQ effects may be of interest to the reader, these effects have both been previously reported in the literature (Eppinger et al., 2013; Schad et al., 2014). We now report these findings in the main Results section and discuss them in more detail in the Discussion section.

Below are direct responses to all reviewer comments, including those summarized above.

*1) I've spent a substantial amount of time mulling over this paper, which really does have many strengths and is exemplary in many ways: a well-motivated task; applied to a large population; combined with an interesting methodology that makes an important step forwards in terms of relating neurobiological/cognitive mechanisms to psychopathology. The results are initially very intriguing* –

*particularly those from the elastic net where impairments in goal-directed control seem to pick out symptoms of compulsivity and intrusive thoughts. However, on reading it more closely there are some important drawbacks which I think require the conclusions to be very significantly toned down; or additional analyses to substantiate them. This, in turn, might make it, in my view, more appropriate for less general journal. My major concerns are: Age has much more of an effect than compulsivity* –

*but OCD prevalence does not increase with age (e.g. Kessler et al. 2005, Arch Gen Psych 62(5):593-602). How can this be if goal-directed impairments underlie compulsivity in such a specific manner? The answer presumably is that goal-directedness depends on multiple processes, and that those related to compulsivity and age might in some way be dissociable. But does that not then make compulsivity a less specific guide to the underlying neurobiology? Isn't this also suggested by the fact that the relationship is, overall, quite weak: in the elastic net, the cross-validated correlation is 0.11? The temporal evolution of OCD decreasing with age also jars with the influence of age.*

We apologize that we believe our presentation caused some confusion here. Age did not have more of an effect on model-based learning than compulsivity (e.g. βAge = -0.049 vs. βCompulsivity = -0.046, difference not significant at p=.83 from mixed effects model), they were statistically equivalent. Nonetheless, the point regarding specificity is an important one and we have clarified our stance in the paper and toned down some of our language to make clear what we mean by ‘specificity’. It was not our intention to suggest that goal-directed deficits are exclusively characteristic or solely causal of compulsivity, and indeed previously published studies have already shown that age, gender and IQ relate to performance on this task (Eppinger et al., 2013; Schad et al., 2014). Our experiment was designed to test if, taking into account these other potentially confounding factors, goal- directed deficits are specific to one well-delineated aspect of psychopathology (e.g., compulsive symptoms or some other construct with some larger scope) compared to others (i.e. depressive or anxious symptoms). This is an important question because in psychiatry, deficits on cognitive tasks are generally found to be non-specific, i.e. observed in multiple diagnosed patient groups.

The reviewers’ point regarding the temporal evolution of OCD is interesting. Indeed, consistent with Kessler et al., OCD symptoms also lessen with age in our sample (as do scores for all psychopathologies). We now mention this in the Discussion and weave it in with our existing discussion regarding multiple processes that may contribute to model-based learning.

We also now report the relationship between psychopathology scores and age, gender and IQ ([Supplementary-material SD1-data]).

*OCD, addiction, etc. are characterised by the positive presence of certain behaviors that bear the hallmark of 'habits'. Why does this not show up in the task? The possibility that the task seems to be insensitive to habitual variation (it never seems to show up in correlations despite the model-free prediction error regressors showing the strongest correlations with BOLD, i.e. neurobiology) somewhat questions the strong conclusions about compulsivity being specifically due to an impairment in goal-directed control: subjects could also have an impairment (excess?) in habitual learning (as one might conclude from excessive habitual behavior in e.g. Gillan et al., 2015 AJP), but this doesn't show up in the task because it's not sensitive. This again makes the conclusions they are drawing from the results just too strong. They state that the literature shows that deficits in model-based but not model-free decision-making has been found and cite Voon et al., Mol. Psych. 2014, but that study used the same task, hence not really addressing this point.*

We agree that we cannot completely rule out the possibility that excessive habit learning may also contribute to compulsive behaviors, as the model-free component of the task is evidently not very sensitive to this, which we make clear in the Discussion. (Though note that the model-free portion of the task does detect relationships with Age, IQ, and gender in the current study, suggesting it is not entirely powerless to detect individual differences in a sample of this size.) However, based on converging data from prior studies, we believe that the most parsimonious explanation for devaluation failures in OCD is deficits in goal-directed control.

The data from the Gillan et al. (2015) paper showed that OCD patients perform habitually in a devaluation test, but this does not provide preferential evidence for excess in habit learning over deficits in goal directed control – this is ambiguous in all devaluation tests. Crucially, the study showed that devaluation failures were associated with dysfunction in a goal-directed structure (the caudate), not for example the putamen, which has been associated with stimulus-response habit learning. While we acknowledge that this is indirect evidence, it converges on the notion that it is goal-directed deficits that are associated with excessive habit-forming in OCD patients. We highlight this observation in the Discussion and also comment on the need for future work to develop a viable marker of individual differences in slow stimulus-response learning that is not confounded with goal- directed control.

*They make statements about patient populations but include neither patients nor any other measure by which functional impairment could be judged, and refer to diagnostic categories ('OCD', 'Alcohol addiction') despite not performing any diagnostic tests. The results, figures and discussion needs to avoid reference to diagnostic categories, and I find the term 'trans-diagnostic' difficult in the absence of any diagnosis. At the core if this is that it is unclear whether the results are driven by what one might observe in a typical patient population. One way to address this is to recapitulate the results only amongst those subjects with scores above cutoffs in any one measure, and then talk about 'putative patients' or so.*

We thank the reviewer for this important point, which we have addressed in two key ways. First, we have carefully omitted any sign of claims of this sort from the paper. While we do of course hope that our results extend to diagnosed patient populations, we acknowledge that we cannot speak to this directly in the present study. We have changed the text throughout to make it clearer that we are talking exclusively about variation of self-report symptoms in the general population; we specifically omit the term “trans-diagnostic”.

For the same reason, we think it is probably inappropriate to even speculate about “putative patients” in the way the reviewer suggests. That said, at the reviewer’s request, we repeated our key analyses using (i) subjects in the top 25% of the population in terms of symptom severity, ‘probably patients’ and (ii) using subjects in the ‘normal range’, i.e. the bottom 75%. Results of these analyses are presented in Figure 10, but we have chosen not to include them in the revised manuscript for precisely the reason stated by the reviewer. We do not know which subjects in this study had psychiatric diagnoses and which did not. Therefore, sub-setting our data to create artificial groups (i.e. ‘putative patients’) is not sensible and may mislead the reader. If the reviewer feels strongly about this point, we are open to including this analysis in the paper, but it is our preference not to.

For the reviewers’ interest, the analysis of ‘putative patients’:

The slopes of the regression lines were broadly consistent across all analyses, such that the relationship between model-based deficits and Factor 2 were observed both in individuals reporting the most severe symptoms and those in the normal range (see Figure 10). In all 9 analyses with individuals in the ‘normal range’, the relationship between Factor 2 and model-based deficits were significant at p<.05. In 5/9 analyses with ‘probably patients’ who were in the top 25% of symptom severity, the relationship between Factor 2 and model-based deficits were significant at p<.05. This analysis had just ¼ of the total sample and was therefore severely underpowered. But nonetheless, the direction and slope of the effect were consistent across the board, providing evidence to suggest that these relationships will likely generalize to patient populations.

Author response image 1.Plotted are the regression lines for 18 different analyses, where the population was split into the top 25% and bottom 75% for each of the nine clinical questionnaires.**DOI:**
http://dx.doi.org/10.7554/eLife.11305.016

To answer a related question (i.e. whether the effects might be driven by discontinuous effects among only the sickest people) we tested the extent to which our effects are truly dimensional, i.e. linear across the range of severity observed in the study, we fitted ‘Factor 2’ including both a quadratic and linear interaction term to our mixed effects model. The quadratic term was not significant (β=-.0016, p=.822), while the linear term was still highly significant (β=-.045, p=.001), indicating that our results can be best described as linear. We include this in the paper (Results, fifth paragraph).

*We also need to know whether those subjects excluded based on performance were typical in terms of self-report.*

In study 1, we found that those subjects who were excluded had lower symptoms of OCD (t(604)=2.477, p=.014), trait anxiety (t(604)=2.225, p=.0265), and a trend towards lower levels of depression (t(604)=1.799, p=.073). These differences were not observed in Study 2, where all questionnaire total scores were equivalent across groups, all p<.05. For both Experiment 1 and 2, results presented in this paper are not altered by the inclusion of these subjects in the analyses (and indeed are slightly stronger when these subjects are included) (Subsection “Exclusion Criteria”, last paragraph).

*There is no information about the stability of the effects over time, and hence the term trait is confusing. In fact, the covariates are mostly measures of state, not trait.*

We agree and have omitted all references to ‘trait’ throughout the manuscript.

*2) On closer inspection, the elastic net analysis is far less convincing than on reading the results* – *the strongest loadings (I tried to sort them in descending order from Table 3):*

*I feel that there are good and bad numbers; Am preoccupied with the thought of having fat on my body; I vomit after I have eaten; I check things more often than necessary; Am terrified about being overweight; Like my stomach to be empty; My heart beats faster than usual.*

*With overall only two items from the OCI-R (the measure of OCD used), and neither of these is being significantly loaded onto by the compulsivity factor.*

We apologize that this was not clear. The reviewer is referring to the last column of [Supplementary-material SD3-data], which are not the betas for the regularized regression – these are the loadings from the FA. The questions are already presented in descending order of importance for the regularized regression in this table, based on the *Beta* column (which is relevant to this analysis). We have taken steps to prevent this kind of confusion in future.

In our previous submission, we reported the FA loadings in the last column of [Supplementary-material SD3-data] to facilitate cross-analysis comparison, but we see now that their prominence in the table could lead the reader to think they were results from the regression. We have changed the ‘loading’ column so that it now no longer reports the numerical loadings on compulsivity, but instead indicates whether or not there was overlap on any of the factors from factor analysis. Specifically, we use F1, F2 or F3 to indicate whether an item in the regularized regression also loaded above a threshold (>.25) onto each of the factors. It should now be clear that of the negative predictors of goal-directed performance, the overlap with Factor 2 (F2) is substantial at 75% (15/20), compared to F2, ‘anxious-depression’ (overlap 10%, 2/20), or F3 ‘social withdrawal’ (overlap 10%, 2/20).

The two items from the OCI-R that the reviewer has flagged above do in fact load substantially on the compulsivity factor, as do all of the OCD items: “I feel that there are good and bad numbers” (loading on ‘compulsivity’ = 0.52), as does “I check things more often than is necessary” (loading on ‘compulsivity’ = 0.47).

We believe there is some misunderstanding here due to another similar formatting problem, for which we apologize. In our previous submission, we provided just a snapshot of the 6 top loadings from each questionnaire in [Supplementary-material SD2-data], but we see that this led the reviewer to erroneously conclude that certain items did not load at all on Factor 2. We apologize for the confusion and now present the results in descending order regardless of questionnaire or origin so that the main contributors to each factor are plainly seen. We believe this makes it clear that all OCD items loaded similarly and very strongly on Factor 2.

*The fact that so many eating disorder items show up certainly deserves some comment beyond it being just another compulsive phenotype, but overall this just doesn't quite capture 'compulsivity'. Only one out of the top 8 items has anything obvious to do with compulsivity (other than referring to a disease which they labelled as compulsive).*

There are four eating disorder items in the regularized regression result and two OCD items. We do not think this difference is particularly notable for several reasons. This difference is likely due to the fact that the eating disorders questionnaire addresses a greater number of putatively discrete DSM disorders compared to the OCD questionnaire. DSM-5 defines OCD as a unitary disorder and the OCI questionnaire records severity of this disorder. The EAT scale addresses at least three different disorders, which the clinician must consider in the context of differential diagnosis: Anorexia Nervosa, Bulimia Nervosa, and Binge Eating Disorder. Moreover, the eating disorder questionnaire also simply has more items (n=26) compared to the OCD questionnaire (n=18) and the addiction scale (n=10).

The suggestion that some disorders are ‘compulsive’ comes from prior work upon which our hypotheses were based. We do not make any original claims about certain disorders being compulsive here, and labeling cannot be construed as circular. As we cite in the introduction, previous work has suggested that OCD, eating disorders and disorders of addiction are characterized by compulsivity, in that patients feel compelled to perform behaviors (i.e. they are urge-driven), which are repetitive and relatively insensitive to negative consequences (e.g. Voon et al., 2014; Godier & Part 2014; Everitt & Robbins, 2005).

Nonetheless, we agree with the reviewers’ more general concern regarding the specificity of the results to *behavioral* compulsivity. Indeed, we did not anticipate that the repetitive thoughts that accompany these compulsions would load so tightly with the behaviors. This is an interesting and novel finding and in response to this and other comments, we have relabeled the factor ‘Compulsive Behavior and Thought’ and discuss this relationship in more detail in the Discussion.

*3) Finally, I do wonder about how overall severity contributes. This is important because severity is strongly related to comorbidity (see e.g. Kessler et al., 2005, in the same volume as above), and hence important for any trans-diagnostic processes. Half the questionnaires are correlated (and picked up by the compulsivity factor). The most severely ill patients might thus be most likely to respond positively on many compulsivity items. Could it be that the most severely impaired patients simply look compulsive because they are more likely to have more comorbid disorders and hence show up in the compulsive category?*

Although an interesting suggestion, a severity hypothesis does not sit with several observations in our study.

Firstly, one of the reasons that assessing the specificity of this effect is so crucial is because it can refute exactly the kind of general severity confound that the reviewer suggests. For example, individuals reporting the most severe depression symptoms in our sample are no more impaired on model-based learning than those with no depression symptoms. In other words, leaving aside the ‘compulsive behavior and thought’ factor for a moment, the basic effects on the total scores of questionnaires show the basic pattern of specificity we later formalize.

Second, depression and anxiety are much more common compared to compulsive disorders (as in Kessler et al., 2005). This means that individuals with more comorbidities would be expected to show depression more often than compulsive symptoms. This again does not sit with the view that the more severe patients are preferentially picked up by the compulsive factor.

Third, although Factor 3 included contributions from fewer questions overall compared to the other factors, factor 1 and 2 had a similar number of loadings. In other words, Factor 2 did not tap into a greater number of symptoms compared to Factor 1.

Fourth, the Mechanical Turk population have been reported to have a significantly higher rates of social anxiety relative to the general population – 7x the 12-month prevalence reported by Kessler et al., 2005 (Shapiro et al., 2013, Clinical Psychological Science). Social anxiety was marginally associated with better goal-directed control, not worse.

*4) In the FA, the first component doesn't contain anxiety at all. Anxiety loads much more on the second factor, and does so possibly even more than compulsivity: there are around 9 or 10 items that clearly relate to anxiety loading onto it, but only 2 items relating to compulsive behaviours.*

We apologize that the way we have displayed the loadings in the [Supplementary-material SD2-data] (much like above for file 2B) was unclear. As described above, we now display the loadings in descending order and we believe this resolves the reviewer’s concern. Anxiety does not load more on Factor 2 than Factor 1. Empirically, of all the items that loaded onto Factor 2 (‘Compulsivity and Related Cognitions’) at >0.25, just 6/87 (7%) are from the STAI-T (trait anxiety inventory), whereas for the Anxious-Depressive factor, 18/79 (23%) are from the STAI-T. Taking into account the total number of STAI-T items available, 90% of these items loaded onto the Anxious-Depressive factor, while just 30% loaded onto the Compulsivity factor. The mean loading of the STAI-T items onto the Anxious-Depressive factor is 0.52, while the mean loading onto compulsivity is 0.15 (difference is significant at p<.001, [Supplementary-material SD2-data]). Finally, the highest average loadings for the Depressive-Anxious factor came from the Trait Anxiety questionnaire (M=0.52, SD=0.17), followed by Apathy (M=0.44, SD=0.16) and Depression (M=0.38, SD=0.23). The data are unequivocal: items from the trait anxiety questionnaire loaded more onto Factor 1 compared to Factor 2.

We now include a more detailed characterization, including many of these descriptive statistics in the Materials and methods section.

To make this clearer, we also now include what was formerly [Supplementary-material SD2-data] as a table in the main manuscript (now Table 2) and we refer the reader there multiple times.

A number of the AUDIT variables are hard to relate to compulsions: alcoholics start drinking early as they experience withdrawal symptoms after a night of sleep. If anything, this component is more related to obsessions, anxious worries and difficulties controlling thoughts – which is, in terms of constructs, much closer to goal-directed deficits, it seems to me.

While we acknowledge that the labeling of Factors is a subjective process, our position is that persistent use of alcohol despite adverse consequences is a compulsive behavior and that all AUDIT items that are indicators of the severity of alcohol addiction and are thus are indicators of compulsion. Experiencing withdrawal is one such indicator, in that the extent to which this is experienced marks the severity of addiction.

Nonetheless, in response to a number of comments received on this issue, we have relabeled this factor as ‘Compulsive Behavior and Thought’. Again, if the appearance of the term “compulsive”, even in this deliberately more inclusive rephrasing remains problematic, we are open to advice.

*5) The task itself isn't obviously specific as it is not clear what the model-free component quite captures. This makes it more of a shame they didn't test components we know impact on m-b choices, such as working memory or stress. Impairments in this are also 'trans-diagnostic', and it would have been nice to show that they don't have the specificity of g-d choices.*

While we cannot rule out a role for stress on model-based learning, we believe that the specificity of our effect to compulsive phenotypes speaks to this in some sense. I.e. the lack of association between ‘Anxious-Depression’ and model-based learning suggests that a general stress mechanism is unlikely. We unfortunately cannot rule out a possible role for working memory in the effects reported, and think this is a plausible hypothesis that warrants further investigation. We highlight this as a target for future research in the Discussion.

*6) Both reviewers expressed concerns about the explanatory power (of excessive habit formation due to deficient model-based control) for understanding clinical aspects of compulsivity. As you outlined in the Introduction, a key motivation for studying the relation between model-based /goal-directed decision-making and compulsive symptoms is the notion that "a deficit in deliberative, goal-directed control may leave individuals vulnerable to rely excessively on forming more rigid habits". I understand why this is a straightforward and attractive perspective to explain certain aspects of compulsivity. However, I think it would also be appropriate to mention challenges and potential limitations of this perspective in the Discussion* – *particularly because the dimensional approach chosen here suggests applicability of the proposed mechanism to clinical phenomena. For example, how exactly would a putative deficit in model-based control lead to prominent symptoms in OCD, such as excessive checking, fear of germs, or desire for order?*

This is an important issue and we now include a paragraph detailing our speculations on this issue and refer the reader to a more detailed exposition published previously (Gillan and Robbins, 2014), along with more recent data illustrating how habits of thought might arise via a similar mechanism (Discussion, seventh paragraph).

*The nomenclature and with it the framing need quite some work, e.g. categorical/dimensional measures, in terms of state/trait distinction, and distinctions between compulsions and obsessions.*

We have addressed each of these points in earlier responses. We make clear that we are dealing with dimensional measures, that we do not have any particular stance or evidence for state vs. trait-dependence of these symptom dimensions and include more detailed commentary on our findings regarding the tight relationship between repetitive compulsive behaviors and the associated cognitions.

*7) The paper is very well written and of beautiful simplicity* – *a pleasure to read. However, sometimes a few more technical details or conceptual distinctions may have to be included in the main text to avoid confusion. First, the Introduction repeatedly refers to unspecified "OCD symptoms" which I found confusing, given that the paper is about the general population and that numerous symptoms of OCD exist. I would recommend avoiding the clinical label OCD and referring to compulsivity instead, stating the specific questionnaire you used. Similarly, in the Results section (second paragraph), there is a tension between using trait labels (impulsivity, compulsivity) and diagnostic labels (eating disorders, alcohol addiction); the latter is confusing (and not quite appropriate), given that your study examines the general population. You could eliminate this tension and, at the same time, increase clarity by always referring to the scores of the respective questionnaires.*

Given there are nine different questionnaires, with acronyms, we feel it is still beneficial to refer for example to ‘eating disorders’ rather than EAT-26. But we now do so in longer form at every instance to eliminate any possibility of confusion, while still allowing the reader to easily digest the material e.g. “total scores on self-report measures of eating disorder severity, impulsivity and alcohol addiction”.

*Second, the Results section should define the measure of model-based learning used (first paragraph). Until I went through the Methods section, I was not sure how exactly model-based learning was operationalised, and whether you were referring to a behavioural readout or to the parameter estimates of a computational model.*

We have made this clearer upfront, and also refer the reader to the Materials and methods section for a full description.

*8) You report analyses based on behavioural readouts (trial-by-trial stay/switch behaviour), not model parameter estimates, because the qualitative conclusions drawn from both types of analyses seemed to be almost equivalent. Does this also hold with regard to how well questionnaire scores can be predicted, or does the computational model have a competitive advantage there? It would be instructive for the technically interested reader if you could include estimates of predictive accuracy for both approaches, perhaps in the supplementary material.*

The correlation between the two estimates of model-based learning is r=.87. The difference between the two approaches in terms of capturing psychopathology is therefore necessarily negligible and we therefore do not think it is worthwhile to include this is the manuscript (except for reporting this correlation coefficient: subsection “Supervised Analysis”). We are fine with being overruled on this, but it is our opinion that this analysis does not make sense given the high correlation.

The requested analysis: we tested the extent to which total scores on all 9 questionnaires and the three factors from our factor analysis could be predicted by model-based learning from the one trial back regression versus the full computational model. Prior to conducting these analyses, we regressed out the effects of age, gender and IQ so that we could directly compare the r_2_ of the models. The difference between the two approaches is negligible. However, the computational model did produce nominally higher r_2_ and lower *p*-values for the relationship between clinical scores and model-based learning.

Author response table 1.Each row reflects the results from an independent analysis where each questionnaire total score (z-transformed) was entered as SymptomScorez in the following model: lm(SymptomScorez ~ Agez + Genderz+ IQz + ModelBasedScore). ModelBasedScore was derived from from the one-trial back regression (first three columns) or the computational model (last three columns results). For each, positive β values indicate that the ModelBasedScore is associated with fewer symptoms, whereas negative β values indicate that the symptom score is associated with increased symptoms.**DOI:**
http://dx.doi.org/10.7554/eLife.11305.017***One-Trial Back Regression***Computational model**Clinical Scores**
β (SE)*p*-value*R^2^*β (SE)*p*-value*R^2^*Eating Disorders**-0.09(0.03)****<.001*******.042****-0.09(0.03)****<.001*******.043**Impulsivity**-0.09(0.03)****.002******.028****-0.10(0.03)****<.001*******.032**OCD**-0.07(0.03)****.012*****.050****-0.07(0.03)****0.005******.051**Alcohol Addiction**-0.06(0.03)****.029*****.052****-0.06(0.03)****.028*****.052**Schizotypy-0.04(0.03).101.028**-0.05(0.03)****.044*****.029**Depression-0.03(0.03).351.031-0.05 (0.03).09.033Trait Anxiety-0.02(0.03).552.038-0.03(0.03).221.038Apathy-0.00(0.03).897.015-0.02(0.03).584.015Social Anxiety0.01(0.03).593.0280.01(0.03).775.028Anxious-Depression-0.00(0.03).967.018-0.02 (0.03).474.018Compulsive Behavior and Thought**-0.11(0.03)****<.001*******.088****-0.12(0.03)****<.001*******.089**Social Withdrawal0.03(0.03).282.0360.02(0.03).440.036

9) In the subsection “Quantifying Model-based Learning (Logistic Regression)”, second paragraph: The significant main effect of Transition is very interesting. Could you please state the direction of this effect and perhaps even offer a (speculative) interpretation? This is another place in which a more thorough analysis of the factors on both sides (task measures and demographic/clinical variables) can be useful.

We now state the direction of this effect and state that this is likely due to small biases due to un-modeled structure in the data that is more correctly captured in the full model fit.